# Nutritional redundancy in the human diet and its application in phenotype association studies

Xu-Wen Wang [1], Yang Hu[2], Giulia Menichetti [1,3], Francine Grodstein[4], Shilpa N. Bhupathiraju[1,2], Qi Sun [1,2,5], Xuehong Zhang [1,2], Frank B. Hu[1,2,5], Scott T. Weiss [1] & Yang-Yu Liu [1,6] ✉

Studying human dietary intake may help us identify effective measures to treat or prevent many chronic diseases whose natural histories are influenced by nutritional factors. Here, by examining five cohorts with dietary intake data collected on different time scales, we show that the food intake profile varies substantially across individuals and over time, while the nutritional intake profile appears fairly stable. We refer to this phenomenon as 'nutritional redundancy' and attribute it to the nested structure of the food-nutrient network. This network enables us to quantify the level of nutritional redundancy for each diet assessment of any individual. Interestingly, this nutritional redundancy measure does not strongly correlate with any classical healthy diet scores, but its performance in predicting healthy aging shows comparable strength. Moreover, after adjusting for age, we find that a high nutritional redundancy is associated with lower risks of cardiovascular disease and type 2 diabetes.

Human dietary intake fundamentally affects our nutrition, energy supply, and health. A better understanding of dietary patterns can help us identify measures to prevent or treat health conditions and diseases such as obesity[1,2], type 2 diabetes[3,4], and cardiovascular disease[5,6]. Indeed, randomized trials have established the benefits[7] of the Mediterranean diet on clinical cardiovascular disease[8] and the Dietary Approaches to Stop Hypertension (DASH)[9] diet on blood pressure control. To improve the diet quality, the 2020-2025 Dietary Guidelines for Americans recommended that consumers should select nutrient-dense foods and beverages, which can provide vitamins, minerals and other health-promoting components and have no or little added sugars, saturated fat, and sodium[10]. For example, intake of whole grains, legumes and vegetables and fruit is recommended to reduce ingredients, such as unhealthy fats, and excess sugar and sodium,

which have been associated with many diseases such as coronary heart disease[11–13] and obesity, liver and other metabolic diseases[14–16].

Food is a complex combination of components that can be classified into nutrients and non-nutrients[17]. The preliminary step to developing a guide to healthy dietary guidance is nutrient profiling[18,19]—the science of understanding which nutrients are present in a given food item. This has many potential applications, ranging from dietary guidance to nutrient labeling and the regulation of health claims[18]. The current nutrient profiling relies on some existing databases, such as the USDA's Food and Nutrient Database for Dietary Studies (FNDDS)[20,21], Frida[22], FooDB[23], and many other databases[24–26] (see Supplementary Section 1.1 for details). Those databases represent the most comprehensive efforts so far to integrate food composition data from specialized databases and experimental data.

[1]Channing Division of Network Medicine, Department of Medicine, Brigham and Women's Hospital and Harvard Medical School, Boston, MA 02115, USA. [2]Department of Nutrition, Harvard T.H. Chan School of Public Health, Boston, MA 02115, USA. [3]Network Science Institute, Department of Physics, Northeastern University, Boston, MA 02115, USA. [4]Rush Alzheimer's Disease Center, Department of Internal Medicine, Rush Medical College, Rush University, Chicago, IL 60612, USA. [5]Department of Epidemiology, Harvard T.H. Chan School of Public Health, Boston, MA 02115, USA. [6]Center for Artificial Intelligence and Modeling, The Carl R. Woese Institute for Genomic Biology, University of Illinois at Urbana-Champaign, Champaign, IL 61801, USA. ✉e-mail: yyl@channing.harvard.edu

Despite the various databases on food constituents, a comprehensive understanding of the food-nutrient relationship remains challenging. Human dietary patterns across the world could be drastically different[27]. For example, the Mediterranean diet is characterized by high consumption of olive oil, legumes and vegetables and lower consumption of non-fish meat products while the Western dietary pattern is characterized by high consumption of highly processed meat[28–30], candy and sweets and low intakes of fruits, and vegetables. Although dietary patterns sometimes exhibit drastically divergent food choices, at a finer resolution, the underlying nutrient palettes could reveal a higher degree of similarity. This implies that despite the variability in food choice and intake, nutrient profiles can be highly stable across individuals and over time.

Here, by analyzing human dietary intake data collected from different cohorts and on different time scales, we show that the food profile varies substantially across individuals and over time, while the nutritional profile is highly stable across individuals and over time. This phenomenon is strongly reminiscent of the phenomenon of functional redundancy observed in the human microbiome, i.e., the taxonomic profile is highly personalized while the gene composition (or functional profile) is highly conserved across individuals[31–33]. Hence, hereafter we will refer to this phenomenon as nutritional redundancy (NR). We find that NR cannot be simply explained by the fact that different foods contain similar nutrients. Instead, it is largely due to more sophisticated features (e.g., the highly nested structure) of the food-nutrient network–a bipartite graph that connects foods to their nutrient constituents. The food-nutrient network also enables us to quantify the level of NR for each diet assessment of any individual, i.e., the personal NR. Interestingly, this personal NR measure does not strongly correlate with any classical healthy diet scores, but its performance in predicting healthy aging shows comparable strength. Moreover, after adjusting for age, we find that a high personal NR is associated with lower risks of cardiovascular disease and type 2 diabetes. Hence, the concept of NR developed here may offer us a new perspective on studying the human diet. The measure of personal NR could serve as a powerful tool in nutrition science to modulate individual dietary patterns and study the correlation with different health phenotypes.

## Results

### The phenomenon of population-level NR

To demonstrate the phenomenon of NR in the human diet, we analyzed five comprehensive datasets with dietary data collected on different time scales. (1) Diet-microbiome association study (DMAS)[34]: a longitudinal study of 34 healthy individuals with dietary intake data and stool samples collected daily over 17 consecutive days. Daily dietary intake data were collected using the automated self-administered 24-h (ASA24) dietary assessment tool[35–37]. DMAS focused on 41 nutrients, and 9 food groups (e.g., grains, fruits, vegetables, etc.) based on the Food and Nutrient Database for Dietary Studies (FNDDS) food coding scheme[20]. Note that there are two outliers in DMAS, referred to as "shake drinkers", whose reported diet consisted primarily of a nutritional meal replacement beverage. Those two outliers were removed in our analysis. Also, in our analysis, we focused on those participants (in total $n = 30$) who have ASA24 data available for all the 17 days. (2) Nurse Healthy Study (NHS)[38,39]: NHS began in 1976 when 121,700 female registered nurses, aged 30–55 years in the United States were enrolled; the semiquantitative food frequency questionnaire (FFQ) was first administrated in 1980, and then administrated approximately every 4 years. So far, we have up to eight time points of FFQ data for NHS participants. In our analysis, we focused on those participants (in total $n = 35,256$) who have FFQ data available for all the eight time points. (3) Health Professionals Follow-up Study (HPFS)[40,41]: HPFS is a prospective cohort of 51,529 men followed since 1986, when participants initially ranged from 40–75 years.

This all-male study was designed to complement the all-female NHS/NHSII, and both were conducted identically. In our analysis, we focused on those participants (in total $n = 17,529$) who have FFQ data available for all the seven time points. (4) Women Lifestyle Validation Study (WLVS)[37]: As a sub-study of the NHS and NHSII, WLVS was designed to investigate the measurement-error structure associated with self-reported dietary and physical activity assessments within 1 year (with up to four ASA24 records). Women with a history of coronary heart disease, stroke, cancer, or major neurological disease were excluded. Among the 796 enrolled participants, 692 completed at least one ASA24. In our analysis, we focused on those WLVS participants with all four ASA24 records available (in total $n = 216$). (5) Men's Lifestyle Validation Study (MLVS)[42,43]: a 1-year study nested in HPFS[40,41]. MLVS was designed to complement WLVS. In our analysis, we focused on those MLVS participants with all of four ASA24 records available (in total $n = 451$).

For those selected participants in each study, we first assessed the change in their food and nutrient profiles (i.e., the relative abundances of food items and nutrients in their diet) over time. The relative abundances of nutrients are calculated by converting the unit of each nutrient to gram and then normalized by the total grams of all nutrients for an individual. We found that the food profiles were highly dynamic for almost all individuals at different time scales: daily (Fig. 1a1), monthly (Fig. 1b1, c1), and yearly (Fig. 1d1, e1). Moreover, the food profiles are highly personalized[44], i.e., the intra-individual dissimilarity of the foods consumed over time is significantly lower than the inter-individual dissimilarity of their consumed foods at both single food (Fig. S1a) and food-group (Fig. S1b) level. By contrast, the nutrient profiles, as expected, were highly conserved across different individuals and over the whole study time period for all five studies (Fig. 1a2–e2), and were not highly personalized (Fig. S1c).

We observed that the most abundant food items consumed by individuals are sweetened beverages and vegetables. This is partially due to the fact that the moisture content in these foods is high, as well as their relevance as staple foods in different cuisines[45]. On the other hand, the most abundant nutrients for all participants are carbohydrate, fat, and protein, confirming their key role as dietary intake macronutrients, and representing broad families of chemical compounds. Within the carbohydrate category, sugars and fiber are the driving factors.

To quantify the between-individual difference in food or nutrient profiles, we adopted the notion of beta diversity from community ecology[46]. In particular, we used four different measures (Bray-Curtis dissimilarity, root Jensen-Shannon divergence, Yue-Clayton distance, and negative Spearman Correlation) to quantify the beta diversity. As shown in Fig. S1, the beta diversity of nutritional profiles is significantly lower than that of food profiles at both single food (Fig. S1d) and food-group (Fig. S1e) levels in terms of all the four measures of beta diversity for all the five studies. While the beta diversity captured by food groups[47] is remarkably lower compared to the food-level analysis, we found that it is still significantly higher than the beta diversity of nutritional profiles.

### Definition of personal NR

Consider a pool of $N$ food items, which contains a collection of $M$ nutrients in total. The food profile $\boldsymbol{f}^{(\nu)} = [f_1^{(\nu)}, \cdots, f_N^{(\nu)}]$ of individual-$\nu$'s diet assessment can be directly related to its nutrient profile $\boldsymbol{n}^{(\nu)} = [n_1^{(\nu)}, \cdots, n_M^{(\nu)}]$ through the FNN (Fig. 2). Here, $f_i^{(\nu)}$ (or $n_a^{(\nu)}$) represents the relative abundance of food-$i$ (or nutrient-$a$) in the diet assessment of individual-$\nu$. We define the FNN as a weighted bipartite graph connecting these foods to their nutrients[5]. The FNN can be represented by an $N \times M$ incidence matrix $\boldsymbol{G} = (G_{ia})$, where a non-negative value $G_{ia}$ indicates the amount contributed by food-$i$ to nutrient-$a$ (see Fig. 2a for the unit of each nutrient). The nutrient

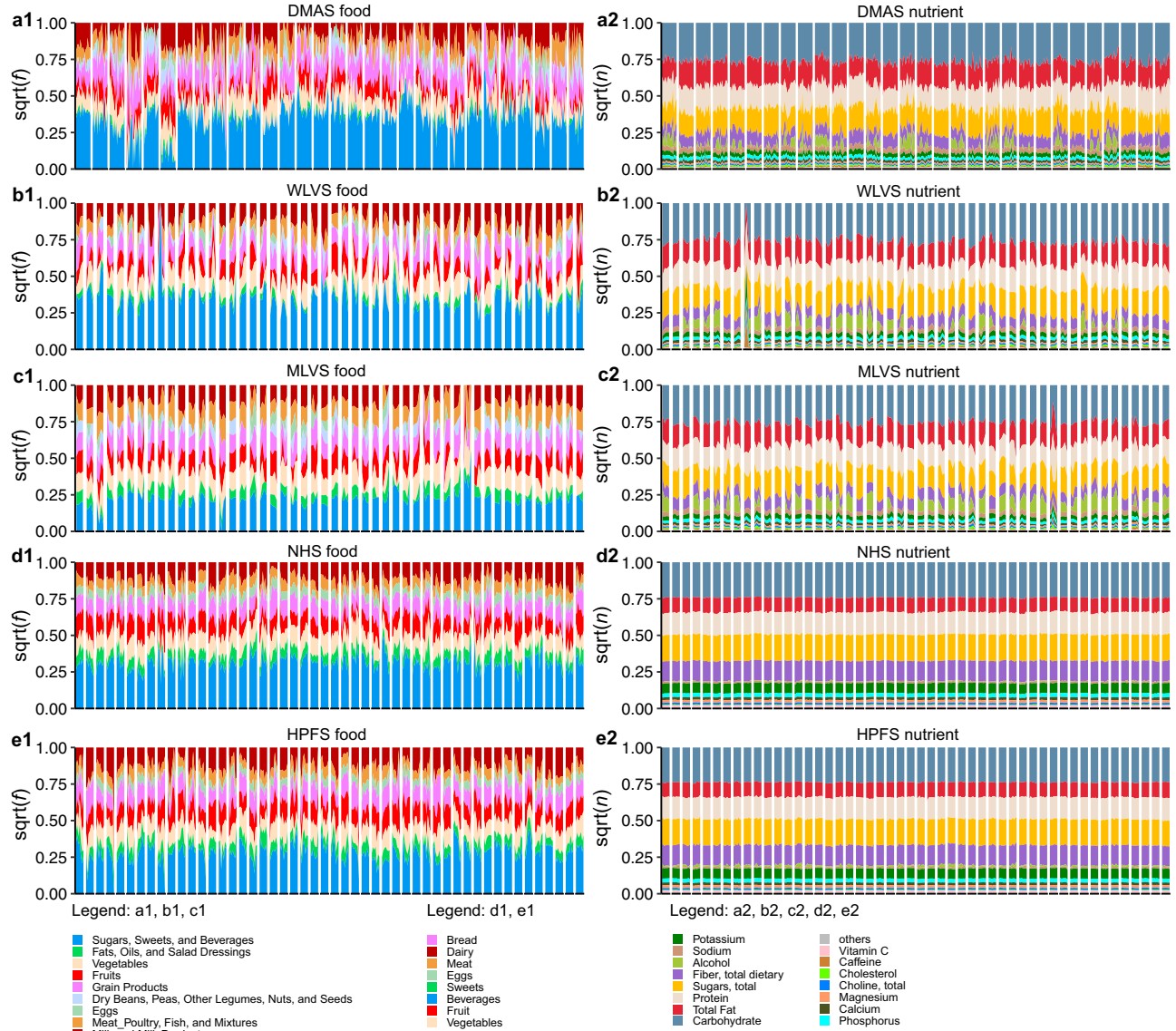

**Fig. 1 | Dietary food profiles are highly personalized while the nutrient profiles are highly stable.** Rows represent results from selected participants of different cohorts: (**a**) DMAS (dietary intake data collected using ASA24 dietary assessment tool daily over 17 consecutive days); (**b**) WLVS (with four ASA24 records within 1 year); (**c**) MLVS (with four ASA24 records within 1 year); (**d**) NHS (with FFQ administrated every 4 years and with total eight time points). **e** HPFS (with FFQ administrated every 4 years and with total seven time points). Columns: (**1**) food profiles;

(**2**) nutrient profiles. For DMAS, we plot the food and nutrient profiles of 30 participants who have ASA24 data available for all the 17 days. For the WLVS, MLVS, and NHS datasets, we plot the food and nutrient profiles of 50 randomly chosen individuals. In the visualization of nutrient profiles, we only show the top-15 most abundant nutrients, while the remaining nutrients (after excluding amino acids, total fatty acids of saturated, monounsaturated, and polyunsaturated) are summarized as others.

profile is given by $\boldsymbol{n}^{(\nu)} = c\boldsymbol{f}^{(\nu)}\boldsymbol{G}$, where $c = \left[\sum_{a=1}^{M}\sum_{i=1}^{N} f_i^{(\nu)} G_{ia}\right]^{-1}$ is a normalization constant.

A key advantage of the FNN is that it enables us to calculate the NR for each diet assessment of any individual, i.e., the within-individual or personal NR. In the ecological literature[32,33,48], the functional redundancy of a local community is interpreted as the part of its taxonomic diversity that cannot be explained by its functional diversity. Similarly, we can define the NR of a dietary assessment from a particular individual as the part of its food diversity (FD) that cannot be explained by its nutrient diversity (ND), i.e., NR = FD − ND. Here we chose FD to be the Gini-Simpson index: GSI $\equiv 1 - \sum_{i=1}^{N} f_i^2 = \sum_{i=1}^{N}\sum_{j\neq i}^{N} f_i f_j$, representing the probability that randomly chosen two units of an individual's food profile (with replacement) belong to two different food groups; and ND is chosen to be the Rao's quadratic entropy $Q \equiv \sum_{i=1}^{N}\sum_{j\neq i}^{N} d_{ij} f_i f_j$, characterizing the mean nutritional distance between any two randomly chosen food items in the diet

assessment[25,26]. Here $d_{ij} = d_{ji} \in [0,1]$ denotes the nutritional distance between food-$i$ and food-$j$. By definition, $d_{ii} = 0$ for $i = 1, \cdots, N$. For the sake of simplicity, to avoid major effects driven by the several orders of magnitude covered by nutrient amount in food[49], we compute $d_{ij}$ as the (unweighted) Jaccard distance between the sets of nutrients within two food items (see Methods for definition). Note that with FD = GSI and ND = Q, we have NR $= \sum_{i=1}^{N}\sum_{j\neq i}^{N} (1 - d_{ij}) f_i f_j$, naturally representing the nutrient similarity (or overlap) of two randomly chosen food items in any diet assessment.

The personal NR of each diet assessment is closely related to the phenomenon of population-level NR observed over a collection of diet assessments. Let's consider highly personalized food profiles from a population. There are two extreme cases: (i) Each food has its own unique nutrient content (Fig. 2b1), hence $d_{ij} = 1$ for any $i \neq j$. In this case, for each individual we have FD = ND and NR = 0 (representing the lowest level of nutritional redundancy), and the nutrition profiles vary

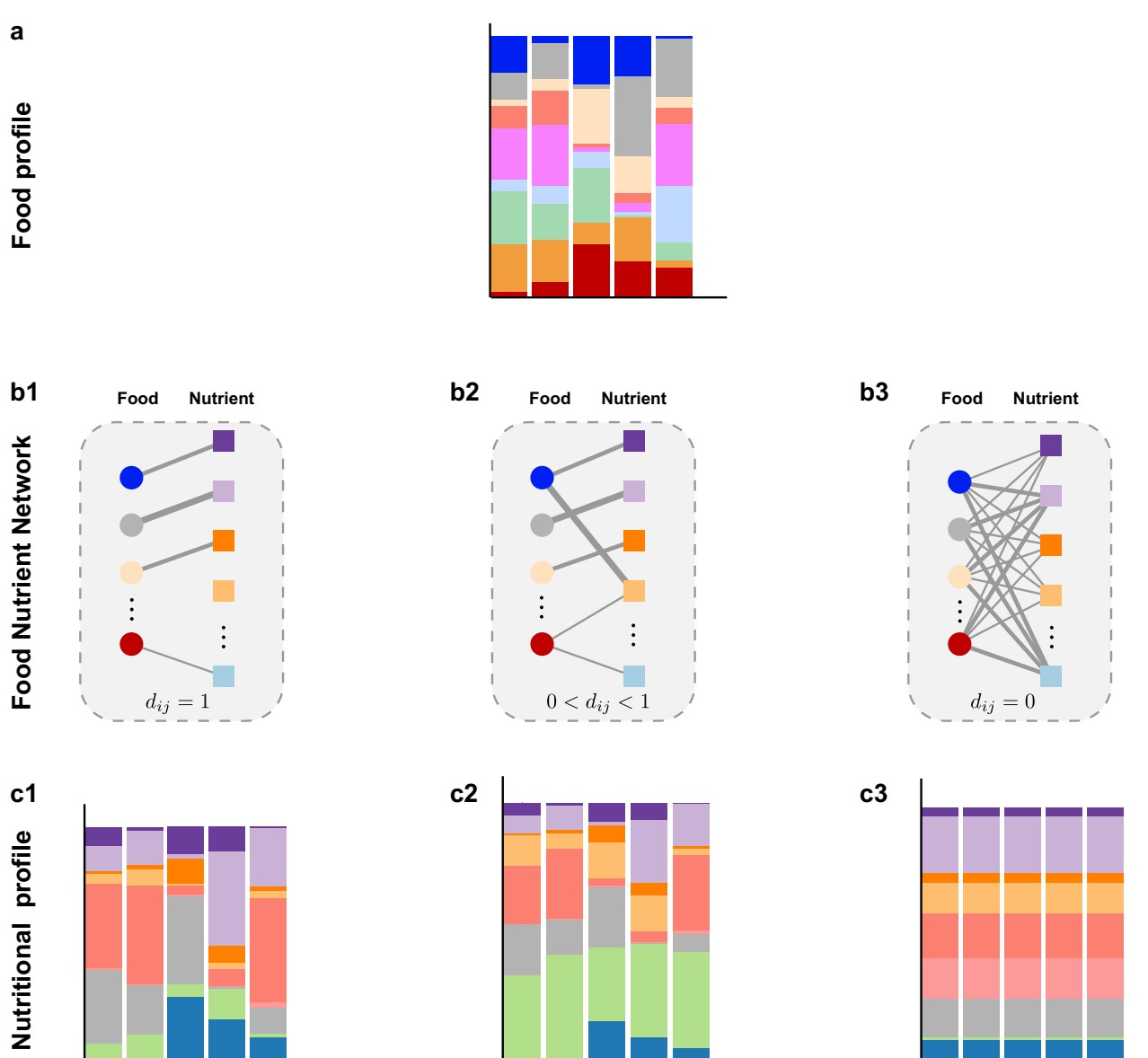

**Fig. 2 | Structure of the food-nutrient network is crucial for determining the nutritional redundancy of human diet. a** The food profiles vary drastically across individuals. **b1–b3** Food-nutrient network is a bipartite graph that connect food items to their nutrient constituents, with edge weight representing the amount contributed by food-$i$ to nutrient-$a$. **b1** Each food has a unique set of nutrient constituents. **b2** Different foods share a few common nutrients; some foods are specialized to have some unique nutrients. **b3** All foods share exactly the same set of nutrients. For each individual, its nutritional profile can be calculated from its food profile in (**a**) and the food-nutrient network in (**b**). **c1** The nutritional profiles vary drastically across different individuals. **c2** The nutritional profiles are highly conserved across different individuals. **c3** The nutritional profiles are exactly the same across all individuals.

drastically across different individuals (Fig. 2c1). (ii) All food share exactly the same nutrient contents (Fig. 2b3), rendering $d_{ij} = 0$ for all $i$ and $j$. In this case, for each individual we have ND = 0 and NR = FD (representing the highest level of nutritional redundancy), and the nutrition profiles are exactly the same for all individuals (Fig. 2c3). These two extreme scenarios are of course unrealistic. In a more realistic intermediate scenario, the FNN has certain topological features such that different foods share a few common nutrients, but some foods are specialized to include some unique nutrients (Fig. 2b2). In this case, the ND and NR of each individual's diet assessment can both be quite comparable, and the nutritional profiles can be highly conserved across individuals (Fig. 2c2).

**A reference FNN**

To visualize the topological features of real FNNs, we constructed a reference FNN based on USDA's Food and Nutrient Database for Dietary Studies (FNDDS) 2011–2012[20], consisting of 7618 foods and 65 nutrients and micronutrients. This reference FNN is depicted as a bipartite graph, where for visualization purposes, each food node represents one of the nine highest-level food groups (based on the FNDDS food coding scheme) and each nutrient node represents nutrient (Fig. 3a). Note that the FNN associated with the diet assessment of any individual can be considered as a particular subgraph of this reference FNN.

To characterize the structure of this reference FNN, we systematically analyzed its network properties using the complete nutrient profile of all 7618 foods. We first visualized its incidence matrix (Fig. 3b), where the presence (or absence) of a link connecting a food and a nutrient is colored in green (or white), respectively. We found that some foods (e.g., bacon cheeseburger, hot ham and cheese sandwich, corresponding to the leftmost columns in Fig. 3b) contribute to almost all of the nutrients, while some foods only include very few nutrients (e.g.,

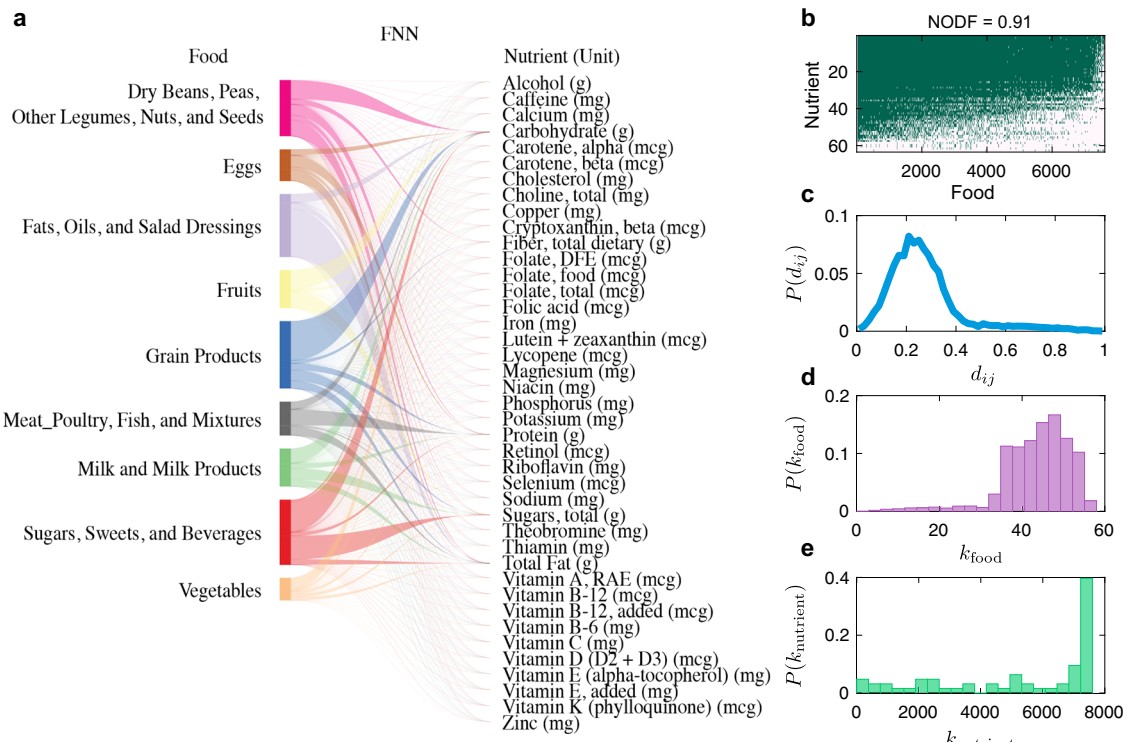

**Fig. 3 | The food-nutrient network (FNN) has a nested structure. a** We used USDA's Food and Nutrient Database for Dietary Studies (FNDDS) to construct the reference FNN, which consists of 7618 foods and 65 nutrients and micronutrients. This reference FNN is depicted as a bipartite graph, where for visualization purposes each food node represents one of the nine highest-level food groups (based on the FNDDS food coding scheme) and each nutrient node represents a nutrient. **b** The incidence matrix of the FNN, where the presence (or absence) of a link between a food and a nutrient is colored in green (or white), respectively. NODF: Nestedness metric based on Overlap and Decreasing Fill. **c** The probability distribution of nutrient distances ($d_{ij}$) among different food items. The bin size is 0.02. **d** The food degree distribution. Here, the degree of a food item is the number of distinct nutrients it contains. **e** The nutrient degree distribution. Here, the degree of a nutrient is the number of food items that contain this nutrient. The 22 nutrients represent the amino acids and total fatty acids of saturated (S) monounsaturated (M), and polyunsaturated (P) were not shown for visualization in (**a**).

sugar substitutes and smart water, corresponding to the rightmost columns in Fig. 3b). Moreover, we noticed that the incidence matrix displays a highly nested structure, i.e., the nutrients of those food items (with fewer distinct nutrients) in the right columns tend to be subsets of nutrients for those food items (with more distinct nutrients) in the left columns. The nestedness of the FNN can be quantified by utilizing the classical Nestedness metric based on Overlap and Decreasing Fill (NODF) measure[50,51], and turns out to be much higher than expected by chance (see Methods for details). We then calculated the nutritional distances among different food items, finding a unimodal distribution with the peak centered around 0.25, indicating that most food items include very similar nutrient components (Fig. 3c). Finally, we calculated the degree distributions of nutrient nodes and food nodes, respectively. Here, the degree of a nutrient node in the FNN is just the number of distinct foods that contain this nutrient. Similarly, the degree of a food node in the FNN is the number of distinct nutrients it contains. We found that the degrees of food items follow a Poisson-like distribution (Fig. 3d), implying that different foods generally contain a very similar number of nutrients. Nutrient degrees show a more extreme behavior, exhibiting a probability density peak located at the higher end of the degree spectrum, indicating that the majority of nutrients are different from zero in almost all foods collected in FNDDS, except a few exceptions (Fig. 3e).

FNN associated with each of the nine food groups and FNNs constructed by using other databases, e.g., Frida[22] and Harvard food composition database (HFDB)[39], revealed very similar network properties (see Figs. S2, S3). This confirms that the nested structure of the FNN is a universal property across different databases, rather than determined by the nutrients' overlap in USDA reference. In addition,

we analyzed the FNN of 51 raw foods in HFDB, finding that the network still displays high nestedness structure. However, NODF of this raw FNN is 0.573, which is much lower than the original FNN (see Fig. S4).

We emphasize that the highly nested structure of the reference FNN is neither explained by the presence of macronutrients, i.e., broad classes of chemicals such as carbohydrates, protein, and fats that are key components of food and exhibit high degree, nor by the nutrient ontology used to annotate the databases. First, as shown in Fig. 3b, the incidence matrix of the FNN still displays a highly nested structure even in the absence of high degree nutrients (the topmost green rows). Second, FNN still shows highly nested structure after excluding nutrients without InChIKey[52], effectively removing the first hierarchical level of the nutrient ontology, and also all those nutrients that correspond to non-specific chemical mixtures (see Fig. S5). Third, if we randomize the FNN but preserve the nutrient degree distribution, the randomized FNNs have much lower nestedness than that of the real FNN, and the nutrient distances between different foods are significantly increased (Fig. S6). Last but not least, we adopted tools from statistical physics[53] to calculate the expected nestedness value and its standard deviation for an ensemble of randomized FNNs in which the expected food and nutrient degree distributions match those of the real FNN. We found that the expected nestedness of randomized FNNs is significantly lower than that of the real FNN (one sample z-test yields $p_{value} < 10^{-5}$, see Methods for details).

**Personal NR calculation based on dietary intake data**

We calculated the personal NR of diet assessments of those selected participants in the four studies: DMAS, WLVS, MLVS, NHS, and HPFS. First, we constructed a reference FNN based on the FNDDS to calculate

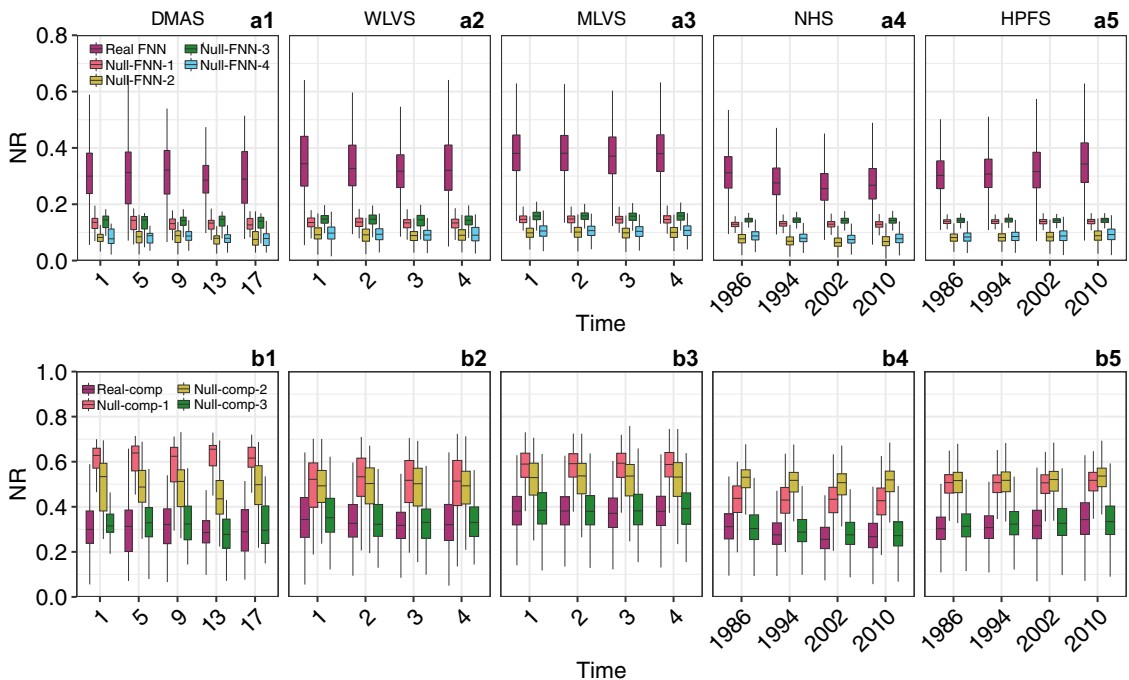

**Fig. 4 | Topological features of the food-nutrient network and the human dietary pattern contributes to the nutritional redundancy. a1–a5** The box plots of the nutritional redundancy were calculated from the real FNN (purple box, $n = 30, 216, 451, 35,256$ and $17,529$ independent subjects), as well as the randomized FNNs (other colored boxes, $n = 30, 216, 451, 35,256$ and $17,529$ independent subjects) using four different randomization schemes: Complete randomization (Null-FNN-1); Food-degree preserving randomization (Null-FNN-2); Nutrient-degree preserving randomization (Null-FNN-3); Food- and nutrient-degree preserving randomization (Null-FNN-4). Here the degree of a nutrient is the number of foods that contain it, and the degree of a food is the number of nutrients contained in it. **b1–b5** The box plots of NR were calculated from the real food composition (Real-comp, purple box, $n = 30, 216, 451, 35,256$, and $17,529$ independent subjects), as well

as the randomized food compositions (other colored boxes, $n = 30, 216, 451, 35,256$ and $17,529$ independent subjects) using three different randomization schemes: Randomized food assemblage generated by randomly choosing the same number of food items from the food pool but keeping the food profile unchanged (Null-comp-1); Randomized food abundance profiles through random permutation of non-zero abundance for each participant across different foods (Null-comp-2); Randomized food abundance profiles through random permutation of non-zero abundance for each food across different participants (Null-comp-3). For the visualization purpose, we only showed four time points for each study. Boxes indicate the interquartile range between the first and third quartiles with the central mark inside each box indicating the median. Whiskers extend to the lowest and highest values within 1.5 times the interquartile range.

the NR of DMSA, WLVS, and MLVS participants, and a reference FNN based on HFDB to calculate the NR of NHS and HPFS participants (see Methods for details). Interestingly, we found that in all the 5 studies (DMAS, WLVS, MLVS, NHS, and HPFS) NR ~ 0.3 (Fig. 4, purple boxes), suggesting that nutritional redundancy and nutrient diversity are generally comparable in human diet.

The NR of the NHS at the population level displays a non-monotonic decrease, indicating that the dietary patterns of those participants indeed have been adjusted. To better illustrate such a dietary pattern change, we projected the bipartite food-nutrient network (constructed from HFDB) into the food space, resulting in a food similarity network (see Fig. 5a). In this network, each node represents a food item and a link connecting food item-$i$ and item-$j$ represents the unweighted Jaccard similarity $s_{ij}$ of their nutrient constituents (see Methods). Here, for visualization purposes, only links with $s_{ij} \geq 0.85$ were retained. We found a clear modular structure in the food similarity network, i.e., food items from the same food group form a densely connected cluster or module (see Fig. 5a), which is consistent with previous study that a food network based on the foods' nutritional similarity displays separately clustered around animal-based foods and plant-based foods at first, and fish and meats are separately clustered in animal-based food cluster and grains, fruits, vegetables, nuts are separately clustered in plant-based food cluster[54]. Then, we examined the individual food similarity network of a particular NHS participant with the largest NR reduction from year 1984 to 2010 (see Fig. 5b, c). We found the density of her food similarity network in 1984 is much higher than that in 2010, suggesting that this participant

consumed foods with more overlapping nutrient constituents in 1984 (Fig. 5b). Moreover, we found the most abundant food items in 2010 were water and yogurt, which do not connect with each other, indicating that she chose foods with more distinct nutrient constituents (Fig. 5c).

## Impact of FNN structure on personal NR

To identify key topological features of the FNN that determine the NR, we adopted tools from network science. In particular, we randomized the FNN using three randomization schemes, yielding three null models. Null-FNN-1: Complete randomization. We keep the number of foods ($N$) and number of nutrients ($M$) unchanged, but otherwise completely rewire the links between foods and nutrients. Null-FNN-2: Food-degree preserving randomization. We keep $N, M$, and the degree of each food node unchanged, but selects randomly the nutrients that link to each food. Null-FNN-3: Nutrient-degree preserving randomization. Here, we keep $N, M$, and the degree of each nutrient unchanged, but select randomly the foods that link to each nutrient. Null-FNN-4: Nutrient-degree and food-degree preserving randomization. Here, we keep $N, M$, and the degree of each nutrient and the degree of each food unchanged, but randomly rewire the links between food nodes and nutrient nodes. Then we recalculated the NR for each diet assessment (Fig. 4). We found that for all the cohorts all the four null models yield much lower NR than that of the real FNN (Fig. 4, purple boxes). This suggests that the real FNN must have certain topological features that determine the high NR in the human diet. Analyzing the network properties of those null models (Fig. S6), we found that those

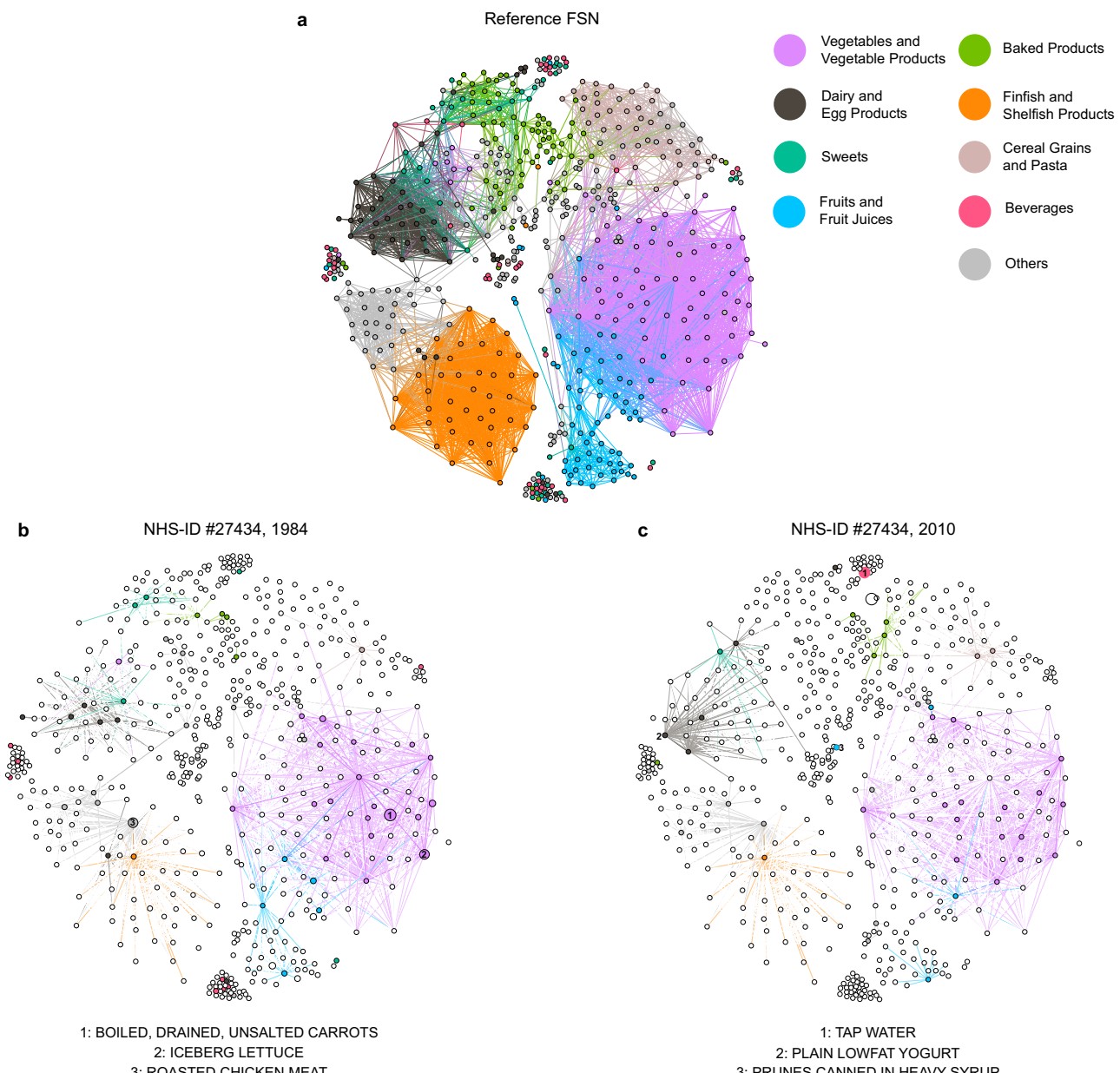

**Fig. 5 | Reference and personal food similarity networks. a** The reference food similarity network (FSN) was projected from the reference food-nutrient network constructed from the Harvard food composition database used in the NHS. The similarity between food-$i$ and food-$j$ was calculated using the unweighted Jaccard similarity index $s_{ij}$. Only links with $s_{ij} \geq 0.85$ were kept in the visualization. The color of each node represents the food group it belongs to. **b** The personal FSN constructed for a particular NHS participant (ID #12137) using her FFQ data collected in 1984. This network is a subgraph of the reference FSN shown in (**a**), thus those nodes and links in global network but do not appear in the subnetwork are shown in white. **c** The personal FSN of the same NHS participant constructed using her FFQ data collected in 2010. In (**b**, **c**), node sizes are proportional to the relative abundance of the food items and edge widths are proportional to the food similarities.

randomized FNNs all display lower nestedness and higher $\langle d_{ij} \rangle$ than those of the real FNN. Thus, the highly nested structure and low $\langle d_{ij} \rangle$ of the real FNN jointly contribute to the NR values observed in the human diet.

### Impact of food composition on personal NR
To test if food composition plays an important role in determining NR, we randomized the food composition of each participant using three different randomization schemes, yielding three different null models. Then we recalculated NR for each participant (Fig. 4b). We found that for each dietary intake record, if we preserve the food composition but randomly replace the food items by those present in the reference FNN, the resulting null model (Null-comp-1) always

yields much higher NR than that of the original dietary record for ASA24 studies but is comparable for NHS and HPFS. This confirms that the food items present in each dietary record are not assembled at random but follow certain assembly rules. Interestingly, if we randomize the food compositions through random permutation of non-zero abundance for each dietary record across different food items (Null-comp-2) or for each food item across dietary records from different participants (Null-comp-3), we found that only Null-comp-3 did not significantly alter NR, suggesting that the assembly rules are consistent across different participants. Taken together, results from the three null composition models suggest that the nutritional redundancy in human diet is low and not randomly selected.

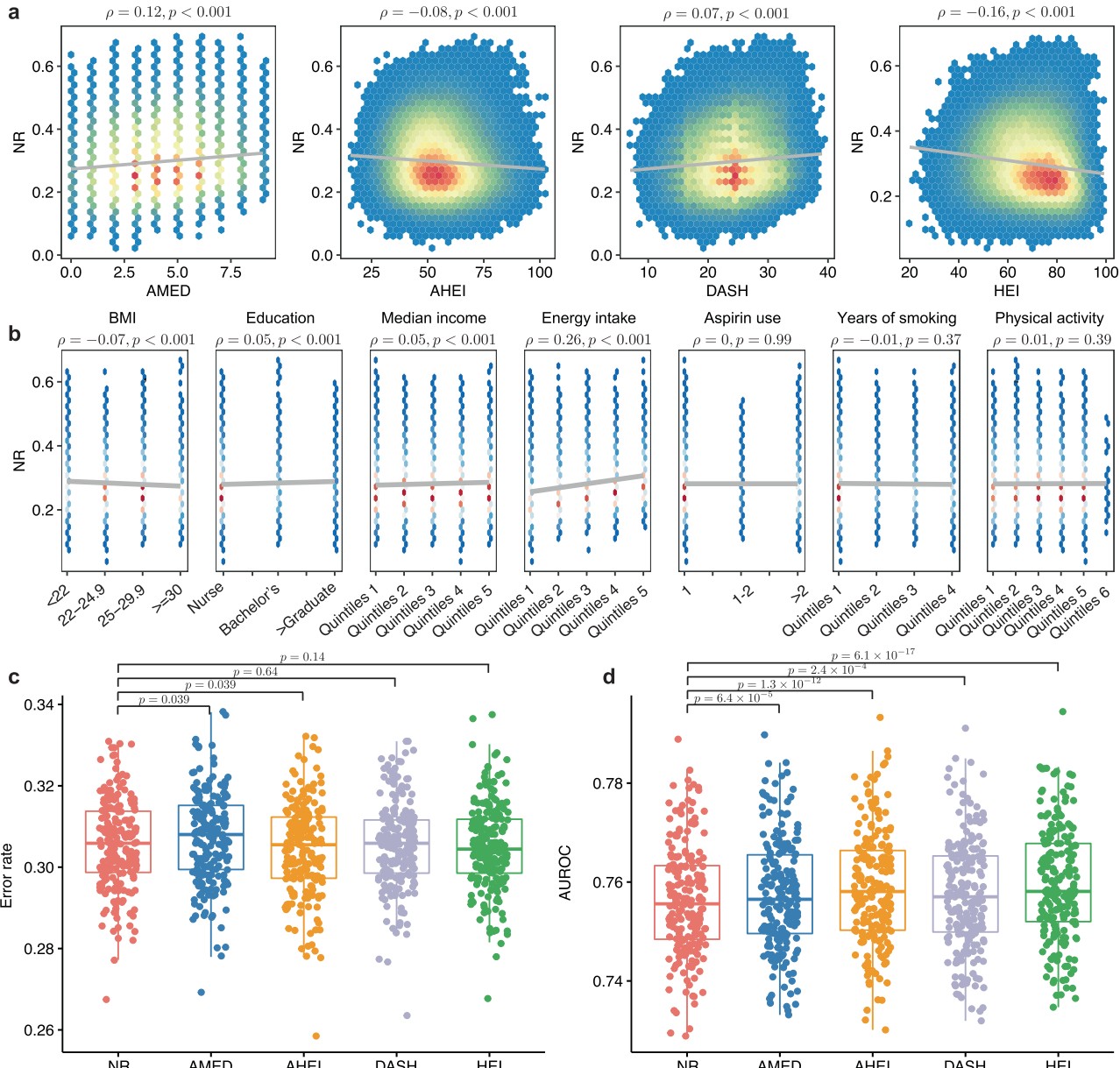

**Fig. 6 | Nutrient redundancy serves as a potential metric to predict healthy aging in NHS. a** Spearman correlation between the nutritional redundancy (NR) and existing healthy diet scores: Alternate Mediterranean Diet Score (AMED); Alternate Healthy Eating Index 2010 (AHEI-2010); Dash Style Diet Score (DASH); and Healthy Eating Index 2005 (HEI-2005). *P* values were calculated from two-sided *t*-test. **b** Spearman correlation between NR and several host factors: body-mass index (BMI); education level; median income; energy intake level; Aspirin use; pack-years of smoking; physical activity. *P* values were calculated from two-sided *t*-test. **c** Error rate of random forest classifier in the prediction of healthy aging

status. **d** AUROC of random forest classifier in prediction of healthy aging status. The participants are randomly spitted into 80% as the training set and the remaining 20% as the test set. The boxplot represents the performances of 200 independent splits. Boxes indicate the interquartile range between the first and third quartiles with the central mark inside each box indicating the median. Whiskers extend to the lowest and highest values within 1.5 times the interquartile range. All FDR-corrected P values were found using the paired and two-sided *t*-test. Significance levels: FDR-corrected $p < 0.0001$(****); <0.01(**); >0.05 (ns).

## Correlation between Personal NR and healthy diet scores

Adherence to healthy diets has the potential to prevent disease and prolong life span. Many healthy diet scores have been developed in the past. To test if the personal NR measure proposed here can be used to quantify a healthy diet, we assessed the association between personal NR and the following four existing healthy diet scores (see Supplementary Section 2 for details): (i) Healthy Eating Index 2005 (HEI-2005)[55]: a score that measures adherence to the USDA 2005 Dietary Guidelines for Americans. (ii) Alternate Healthy Eating Index 2010 (AHEI-2010)[55]: a score that measures adherence to a dietary pattern based on foods and nutrients most predictive of disease risk in the

literature. (iii) Alternate Mediterranean Diet Score (AMED)[56]: a score adapted from form Mediterranean Diet Score of Ref. 57. (iv) DASH Style Diet Score (DASH)[58]: a score capturing the characteristics of the Dietary Approaches to Stop Hypertension diet. As shown in Fig. 6a, we found that the NR is positively correlated with AMED (Spearman correlation coefficient $\rho = 0.12$) and DASH ($\rho = 0.08$), but negatively correlated with HEI ($\rho = -0.16$) and AHEI ($\rho = -0.08$). But in all cases, the Pearson correlation coefficients are weak, implying that personal NR should not be interpreted as being synonymous with a healthy diet score. In other words, a heathy diet does not necessarily have higher or lower nutritional redundancy.

**Table 1 | Hazard ratios (95% confidence intervals) of type 2 diabetes according to tertiles of NR in the Nurses' Health Study (NHS, 1984–2014) and Health Professionals Follow-Up Study (HPFS, 1986–2016)**

|  | T1 | T2 | T3 | P for trend[a] |
|---|---|---|---|---|
| **NHS** |  |  |  |  |
| Cases/Person-year | 1436/312,411 | 1245/312,509 | 1133/312,388 |  |
| Age-adjusted model | 1 (reference) | 0.86 (0.80, 0.93) | 0.78 (0.72, 0.85) | <0.001 |
| Multivariable-adjusted model[b] | 1 (reference) | 0.93 (0.86, 1.01) | 0.93 (0.85, 1.00) | 0.0997 |
| **HPFS** |  |  |  |  |
| Cases/Person-year | 684/147,746 | 502/148,000 | 502/148,085 |  |
| Age-adjusted model | 1 (reference) | 0.73 (0.65, 0.82) | 0.73 (0.65, 0.82) | <0.001 |
| Multivariable-adjusted model[b] | 1 (reference) | 0.77 (0.69, 0.87) | 0.82 (0.73, 0.93) | 0.0016 |

[a]P for trend was calculated using the median value of each tertiles (two-sided Chi-square test).
[b]Multivariable-adjusted model adjusted for age (years), ethnicity (white, African American, Asian, others), body mass index (<21.0, 21.0–22.9, 23.0–24.9, 25.0–26.9, 27.0–29.9, 30.0–32.9, 33.0–34.9, or ≥35.0 kg/m2), smoking status (never smoked, past smoker, currently smoke 1–14 cigarettes per day, 15–24 cigarettes per day, or ≥25 cigarettes per day), alcohol intake (0, 0.1–4.9, 5.0–9.9, 10.0–14.9, 15.0–29.9, and ≥30.0 g/d), hypertension (yes, no), hypercholesterinemia (yes, no), multivitamin use (yes, no), physical activity (quintiles), alternative healthy eating index, family history of diabetes. In NHS, postmenopausal hormone use (never, former, or current hormone use, or missing) and oral contraceptive use were additionally adjusted.

## Personal NR as an indicator of heathy aging

Then, we asked whether personal NR can be an indicator of healthy aging–an overall indicator we developed by combining measures of physical function, cognitive function, mental health, and chronic diseases[59–61]. To address this question, we focused on a subset of NHS participants (n = 21,299) for whom we had all of these four healthy aging components: healthy agers (n = 3491) and usual agers (n = 17,808) through a previous study[62]. The healthy agers are defined as those participants who survived beyond 65 years of age, with no history of chronic diseases, no reported memory impairment, no physical disabilities, and intact mental health. The remaining participants who survived but did not achieve good health in one or more domains were usual agers. The chronic disease domain used to define usual agers includes cancer (other than nonmelanoma skin cancer), myocardial infarction, coronary artery bypass surgery or percutaneous transluminal coronary angioplasty, congestive heart failure, stroke, type 2 diabetes mellitus, kidney failure, chronic obstructive pulmonary disease, Parkinson's disease, multiple sclerosis, and amyotrophic lateral sclerosis. Other host factors that were collected in this previous study include age, education (registered nurse, bachelor's degree, master, or doctorate), marital status (widowed, married, and single/separated/divorced), median income from census tract (quintiles), BMI (<22, 22–24.9, 25–29.9, and ≥30 kg/m2), energy intake (quintiles of kcal/day), multivitamin use (yes/no), aspirin use (<1, 1–2, or >2 tablets/week), pack-years of smoking (quintiles), and physical activity (quintiles of MET-h/week) (see Table S1 for details of those characteristics).

We first assessed the correlations between NR and these host factors. We found that BMI and pack-years of smoking were negatively correlated with NR, while education, median income, total energy intake, and physical activity were positively correlated with NR (Fig. 6b). In all cases, the Spearman correlation coefficients are very weak. To compare the prediction performance of each of the four healthy diet scores (HEI-2005, AHEI-2010, AMED, DASH) with that of personal NR in predicting healthy aging, we built a random forest classifier. In particular, we used features (personal NR or one of the healthy diet scores, and the host factors mentioned above) collected for those participants in 1998 to predict their heathy aging status in 2012 (see Supplementary Section 5 for details). We found that personal NR can achieve very similar error rate (i.e., the proportion of participants that have been incorrectly classified by the model) and AUROC (area under the ROC curve) as other four healthy diet scores (Fig. 6c, d). Interestingly, we found NR can serve as a better indicator than many other host factors (see Fig. S7 for the importance ranking of those factors). We emphasized that the healthy aging prediction does not rely on the particular NR definition or classifier. We evaluated the performance of NR in healthy

aging prediction using the Hill number-based definition and another ensemble classifier: XGBoost[63], showing that both AUC and error rate are robust to different Hill numbers and classifier (see Figs. S8, S9 and Methods for details).

We also performed the healthy aging prediction using data from a substudy of HPFS with 6160 healthy agers and 11,534 usual agers[64]. Again, we used personal NR or one of the four healthy diet scores in 1998 and other host factors to predict the healthy aging status. We found that NR can also achieve very similar error rate (or AUROC) as other healthy diet scores in HPFS. Moreover, the performance of NR in HPFS is comparable to that in NHS (Fig. S10).

## The association between personal NR and the risks of type 2 diabetes and cardiovascular disease

To further demonstrate the potential of the personal NR measure in predicting well-defined phenotypes, we examined its associations with the risks of type 2 diabetes and cardiovascular disease in NHS and HPFS (see Supplementary Section 6 for detailed definitions of the two disease outcomes). In particular, for each disease outcome, we used the age (months)- and calendar year-stratified Cox proportional-hazard model to compute the hazard ratios and 95% confidence intervals (CIs) of the disease according to tertiles of the NR for NHS participants from 1984 to 2014, and HPFS participants from 1986 to 2016.

For NHS participants, after adjusting for age, we observed the NR is associated with a lower risk of the type 2 diabetes (see Table 1). In particular, those NHS participants whose NR values are at tertile-2 and tertile-3 have a hazard ratio of 0.86 (95% CI: 0.80–0.93) and 0.78 (95% CI: 0.72–0.85), respectively, with P for trend <0.001. To further check if this association is robust against many other confounding factors, we also adjusted for total energy intake, race (white, African American, Asian, others), BMI (<21.0, 21.0–22.9, 23.0–24.9, 25.0–26.9, 27.0–29.9, 30.0–32.9, 33.0–34.9, or ≥35.0 kg/m2), smoking status (never smoked, past smoker, currently smoke 1–14 cigarettes per day, 15–24 cigarettes per day, or ≥25 cigarettes per day), alcohol intake (0, 0.1–4.9, 5.0–9.9, 10.0–14.9, 15.0–29.9, and ≥30.0 g/d), hypertension (yes, no), hypercholesterinemia (yes, no), multivitamin use (yes, no), physical activity (quintiles), alternative healthy eating index, family history of diabetes, postmenopausal hormone use (never, former, or current hormone use, or missing), and oral contraceptive use. We found that those NHS participants whose NR values are at tertile-2 and tertile-3 have a hazard ratio of 0.93 (95% CI: 0.86–1.01) and 0.93 (95% CI: 0.85–1.00), respectively, with marginal P for trend = 0.09. For cardiovascular disease, we performed the same calculations. Again, we observed that NR is associated with lower risk of the cardiovascular disease for NHS participants (see Table 1). In particular, after adjusting for age

**Table 2 | Hazard ratios (95% confidence intervals) of cardiovascular disease according to tertiles of NR in the Nurses' Health Study (NHS, 1984–2014) and Health Professionals Follow-Up Study (HPFS, 1986–2016)**

|  | T1 | T2 | T3 | P for trend[a] |
|---|---|---|---|---|
| **NHS** |  |  |  |  |
| Cases/Person-year | 1477/325,684 | 1324/325,688 | 1250/325,419 |  |
| Age-adjusted model | 1 (reference) | 0.92 (0.85, 0.99) | 0.85 (0.79, 0.92) | <0.001 |
| Multivariable-adjusted model[b] | 1 (reference) | 0.94 (0.87, 1.02) | 0.90 (0.83, 0.97) | 0.0057 |
| **HPFS** |  |  |  |  |
| Cases/Person-year | 1358/145,486 | 1285/145,662 | 1290/145,726 |  |
| Age-adjusted model | 1 (reference) | 0.94 (0.87, 1.02) | 0.89 (0.83, 0.96) | 0.0041 |
| Multivariable-adjusted model[b] | 1 (reference) | 0.97 (0.89, 1.04) | 0.92 (0.85, 1.00) | 0.0405 |

[a]P for trend was calculated using the median value of each tertiles (two-sided Chi-square test).
[b]Multivariable-adjusted model adjusted for age (years), ethnicity (white, African American, Asian, others), body mass index (<21.0, 21.0–22.9, 23.0–24.9, 25.0–26.9, 27.0–29.9, 30.0–32.9, 33.0–34.9, or ≥35.0 kg/m2), smoking status (never smoked, past smoker, currently smoke 1–14 cigarettes per day, 15–24 cigarettes per day, or ≥25 cigarettes per day), alcohol intake (0, 0.1–4.9, 5.0–9.9, 10.0–14.9, 15.0–29.9, and ≥30.0 g/d), hypertension (yes, no), hypercholesterinemia (yes, no), multivitamin use (yes, no), physical activity (quintiles), alternative healthy eating index, family history of myocardial infarction. In NHS, postmenopausal hormone use (never, former, or current hormone use, or missing) and oral contraceptive use were additionally adjusted.

(months) only, the P for trend <0.001. After adjusting for a wide range of confounding factors, the P for trend = 0.006.

For HPFS participants, we observed similar results. For type 2 diabetes, after adjusting for age (months) only, the P for trend <0.001; after adjusting for a wide range of confounding factors, the P for trend = 0.002 (see Table 1). For cardiovascular disease, after adjusting for age (months) only, the P for trend = 0.004; after adjusting for a wide range of confounding factors, the P for trend = 0.04 (see Table 2).

For both disease outcomes and both cohorts, we also repeated the above calculations using quintiles of the NR score. As shown in Tables S2–S3, we found qualitatively very similar results.

Since NR is a part of FD and actually they are positively correlated (see Fig. S11), we wonder if FD itself is associated with the risk of type 2 diabetes and cardiovascular disease. We performed association analyses. Interestingly, for both NHS and HPFS participants, we found that, after adjusting for a wide range of confounding factors, FD is not associated with lower risk of type 2 diabetes (see Table S4) or cardiovascular disease (see Table S5) at all. This result implies that the association between NR and disease risks cannot be simply attributed to FD.

To understand the association between NR and the risk of type 2 diabetes and cardiovascular disease in the two cohorts, we analyzed the food consumption pattern of each NR tertiles in those two cohorts. We found that there is a consistent trend among the three NR tertiles for those important food groups in NHS and HPFS (see Fig. 7). For instance, abundances of Fruits, Vegetables, Dairy, Cereal Grains are much higher for T3 (i.e., high-NR participants) than T2 and T1; while abundances of Beverages are much lower for T3 than T2 and T1 in both NHS and HPFS. This food consumption pattern might explain why NR is an indicator of low risk of type 2 diabetes and cardiovascular disease.

## Discussion

Through examining various human dietary intake datasets, we found that food profile varies tremendously across individuals and over time, while the nutritional profile is highly conserved across different individuals and over time. To quantify this nutritional redundancy, we constructed the food-nutrient network–a bipartite graph that connects foods to their nutrient constituents. This food-nutrient network also allows us to assess the NR of any dietary assessment from any individual. We found that this personal NR is not strongly correlated with any existing healthy diet scores. We emphasize that, as the difference between the food diversity and the nutrient diversity of a person's dietary assessment, the personal NR quantifies the nutrient similarity (or overlap) of two randomly chosen food items in the diet assessment. Thus, a healthy diet does not

necessarily have higher or lower NR. Interestingly, we found that the personal NR can be used to predict healthy aging with equally strong performance as those healthy diet scores. Hence, the concept of personal NR offers us a completely new perspective on studying human diet. Moreover, we examined its associations with the risks of type 2 diabetes and cardiovascular disease in NHS (all female) and HPFS (all male). For both cohorts, we found a clear inverse association between NR and the two phenotypes after adjusting for age. For HPFS, the inverse association is observed even after adjusting for a wide range of confounding factors. Whether these findings can lead to practical nutritional guidance warrant further interventional studies.

Since the personal NR measure is not strongly correlated with any classical healthy diet scores, in principle we can combine the concepts of NR and those healthy diet scores to better capture the total impact of diet on health outcomes. For instance, one can leverage the food-specific subgraphs of the FNN (see Fig. S2) to calculate the NR of food groups contributing to each component of a healthy diet score. This will enable us to define an NR-aware healthy diet score. Systematically exploring this direction warrant dedicated efforts, which is beyond the scope of the current work.

There are several limitations in our current framework of NR calculation. First, we did not explicitly consider the nutrient difference between different food sources. We understand that nutrient content and its fluctuations span several orders of magnitude, and different scaling transformation, as well as different selections of nutrients, could modulate nutrient diversity and redundancy across individuals[49]. We anticipate that incorporating this information in our NR calculation will further improve the power of using NR to predict healthy aging or other disease risks[65].

Second, the calculation of a personal NR relies on food intake measurements, e.g., ASA24 and FFQ, which are based on self-reported dietary intake questionnaires. We understand that such food intake measurements have inherent limitations, particularly measurement error related to poor recall, which can be overcome by the use of nutritional biomarkers that are capable of objectively assessing food consumption in different biological samples without the bias of self-reported dietary assessment[66]. Although nutritional biomarkers provide a more proximal measure of nutrient status than dietary intake, quantitatively studying NR using nutritional biomarkers is beyond the scope of the current study. We anticipate that our framework will trigger more research activities in this direction.

Third, due to the unmapped chemical complexity of food[26], the nutritional components listed in existing epidemiological databases represent only a small fraction of the several hundreds of thousands of molecules documented in food[67], many of which are bioactive

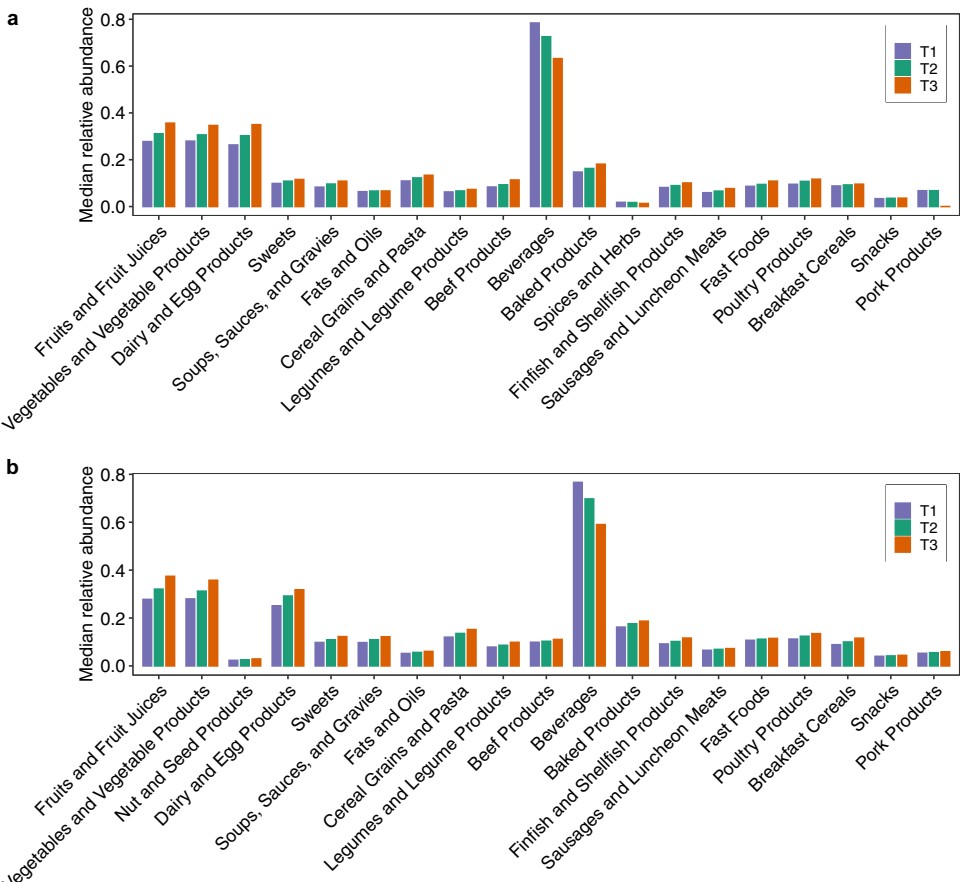

**Fig. 7 | Individuals with high NR tend to consume more healthy food groups.** Square root of median relative abundance of each food group for each tertile group (T1–T3) based on the NR of participants in NHS (**a**) and HPFS (**b**). Food group was defined in Harvard food nutrition database and those food groups presented in less than three tertile groups were not shown.

substances contributed by the extensive secondary metabolism of plants, and substrates for the human gut microbiome[68,69]. Our NR calculation is based on the existing nutrient databases that are certainly incomplete. We anticipate that a more comprehensive food composition database will yield a more powerful measure of NR to predict healthy aging and other disease risks.

Forth, although we considered four null models of the bipartite food-nutrient network, systematically exploring other null models of this bipartite network (or its unipartite presentations) deserves dedicated efforts, which is beyond the scope of the current study. Hence, our analysis cannot entirely exclude the possibility of the presence of other network characteristics that are simpler than nestedness itself and can still be used to explain the NR values observed in the human diet.

Finally, food distance was measured using all nutrient components, which might introduce biases as some nutrients are correlated or redundant, such as fatty acids and amino acids. Although we have demonstrated that the FNN still shows a highly nested structure after excluding broad families of chemicals (Fig. S5), and that removing fat-related nutrient subtypes does not drastically alter NR (Fig. S12), more efforts are warranted to systematically address the potential impact of nutrition ontology on NR calculation.

## Methods
### Food nutrient network
Consider a pool of $N$ food items, which contains a collection of $M$ nutrients in total. The food profile $\boldsymbol{f}^{(\nu)} = [f_1^{(\nu)}, \cdots, f_N^{(\nu)}]$ of the diet record of individual-$\nu$ can be directly related to its nutrient profile $\boldsymbol{n}^{(\nu)} = [n_1^{(\nu)}, \cdots, n_M^{(\nu)}]$ through the FNN (Fig. 2). Here, $f_i^{(\nu)}$ (or $\boldsymbol{n}_a^{(\nu)}$)

represents the relative abundance of food-$i$ (or nutrient-$a$) in the diet record. And we define the FNN as a weighted bipartite graph connecting these foods to their nutrients. The FNN can be represented by an $N \times M$ incidence matrix $\boldsymbol{G} = (G_{ia})$, where a non-negative value $G_{ia}$ indicates the amount contributed by food-$i$ to nutrient-$a$ (see Fig. 2a for the unit of each nutrient). The nutrient profile is given by $\boldsymbol{n}^{(\nu)} = cf^{(\nu)}\boldsymbol{G}$, where $c = [\sum_{a=1}^{M}\sum_{i=1}^{N}f_i^{(\nu)}G_{ia}]^{-1}$ is a normalization constant.

### Nestedness
Nestedness a classical concept in ecology, which is used to characterize the nested structure of ecological systems, such as the species-site network (describing the distribution of species across geographic locations), and the species-species interaction networks (e.g., host-parasite, plant-pollinator interactions)[70–74]. In principle, an ecological system is said to be nested if the items belonging to "smaller" elements (e.g., a small island containing few species, or a specialist species with few interactions) tend to be a subset of the items belonging to "larger" elements (e.g., a large island containing many species, or a generalist species with many interacting partners). Mathematically, those ecological systems can be represented as bipartite graphs with two types of nodes, e.g., sites and species, hosts and parasites, plants and pollinators, etc. In this work, we focus on the food nutrient network of dietary, which is also a bipartite graph with two types of nodes: foods and nutrients.

### Numerical calculation
Consider a general bipartite graph with $N$ type-1 nodes and $M$ type-2 nodes. The structure of this bipartite graph can be represented by its

$N \times M$ binary incidence matrix $\boldsymbol{B} = (B_{ia})$, where $B_{ia} = 1$ if there is a link connecting the $i$-th type-1 node and the $a$-th type-2 node, and 0 otherwise. Mathematically, nestedness can be defined as a property of the incidence matrix $\boldsymbol{B}$. If there exists a permutation of rows and columns such that the set of links in row-$i$ contains the links in row-$(i+1)$, and the set of links in column-$a$ contains those in column-$(a+1)$, then B is a perfectly nested binary matrix. For example, consider the mainland and a series of islands sorted according to their distances to the mainland. The mainland contains all the species, the first island has a subset of species in the mainland, the second island has a subset of species in the first island, etc.

The Nestedness metric based on Overlap and Decreasing Fill (NODF) nestedness metric is based on two simple properties: decreasing fill and paired overlap. For any given bipartite graph with incident matrix $\boldsymbol{B}$, the unweighted degree of $i$-th food node is $k_i = \sum_{a=1}^{M} B_{ia}$, and the unweighted nutrient degree of $a$-th nutrient node is $k_a = \sum_{i=1}^{N} B_{ia}$. The number of nutrients shared by the $i$-th food and the $j$-th food is given by $P_{ij} = \sum_{a=1}^{M} B_{ia}B_{ja}$. And the number of common foods that both of $a$-th nutrient and $b$-th nutrient are included is given by $Q_{ij} = \sum_{i=1}^{N} B_{ia}B_{ib}$. Define $\widetilde{P}_{ij} = 0$ if $k_i = k_j$ and $\widetilde{P}_{ij} = P_{ij}/\min(k_i, k_j)$ otherwise. Then, define $\widetilde{Q}_{ab} = 0$ if $k_a = k_b$ and $\widetilde{P}_{ab} = P_{ab}/\min(k_a, k_b)$ otherwise. The NODF is defined as

$$\text{NODF} = \frac{\sum_{i<j}^{N} \widetilde{P}_{ij} + \sum_{a<b}^{M} \widetilde{Q}_{ab}}{\frac{N(N-1)}{2} + \frac{M(M-1)}{2}}, \tag{1}$$

### Theoretical approach
To theoretically analyze the nested structure of a given bipartite graph, one can construct a grand canonical ensemble for this bipartite graph under the constraint that, for the two types of nodes, the degree sequences in the ensemble match on average the empirical ones[53]. This theoretical approach has two big advantages. First, constraining the ensemble's mean degree sequence to be equivalent to the empirical one limits the possible effects of noisy data, hence possible missing (false negative) or overrated (false positive) links can be dealt with appropriately. Second, for this bipartite graph ensemble one can analytically derive the mean and standard deviation of the distribution of any network property (such as the classical NODF measure of nestedness) that can be analytically formulated in terms of the elements of the bipartite adjacency matrix $\boldsymbol{B}$.

### Nutritional distance measure
To avoid the influence of nutrient amount variability in foods, we used the (unweighted) Jaccard index to quantify the nutritional distance between food item-$i$ and item-$j$:

$$d_{ij} = 1 - \frac{|G_i \cap G_j|}{|G_i \cup G_j|}, \tag{2}$$

where $G_i$ represents the nutrients in food $i$. $d_{ij} = 0$ indicates that the food item-$i$ and food item-$j$ share exactly the same nutrient constituents; $d_{ij} = 1$ means that they have totally different nutrient constituents. The nutritional similarity between food item-$i$ and item-$j$ can be defined as

$$s_{ij} = 1 - d_{ij}. \tag{3}$$

### Nutritional redundancy measure
In the main text, the nutritional redundancy (NR) is defined as: $\text{NR}_\alpha = \text{FD}_\alpha - \text{ND}_\alpha$. $\text{FD}_\alpha$ is chosen to be the Gini-Simpson index:

$\text{GSI} \equiv 1 - \sum_{i=1}^{N} p_i^2 = \sum_{i=1}^{N}\sum_{j\neq i}^{N} p_i p_j$, representing the probability that two randomly chosen members of a subject's food profile (with replacement) belong to two different food items; and $\text{ND}_\alpha$ is chosen to be the Rao's quadratic entropy $Q \equiv \sum_{i=1}^{N}\sum_{j\neq i}^{N} d_{ij} p_i p_j$. Note that with $\text{FD}_\alpha = \text{GSI}$ and $\text{ND}_\alpha = Q$, we have $\text{NR}_\alpha = \sum_{i=1}^{N}\sum_{j\neq i}^{N}(1 - d_{ij})p_i p_j$, naturally representing the nutritional similarity (or overlap) of two randomly chosen members in any subject's food profile. Here, we emphasize that various taxonomic and functional diversity measurements can be used in NR definition, for example Hill number.

### Food diversity
Consider a subject' food profile of $N$ foods with the relative abundance given by a vector $\boldsymbol{f} = [f_1, \cdots, f_N]$. Hill introduced effective *number of species*, which assumes that the taxonomic diversity (of order $q$) of a given subject with relative abundance profile $\boldsymbol{f} = [f_1, \cdots, f_N]$ is the same as that of an idealized subject of $D$ equally abundant foods with relative abundance profile $\widetilde{\boldsymbol{f}} = [1/D, \cdots, 1/D]$.

$$\sum_{i=1}^{N} f_i^q = \sum_{i=1}^{D}\left(\frac{1}{D}\right)^q = D^{1-q}, \tag{4}$$

This offers a parametric class of food diversity measures defined as follows:

$$\text{FD}_q : = \left(\sum_{i=1}^{N} f_i^q\right)^{\frac{1}{(1-q)}} \text{ for } q \neq 1. \tag{5}$$

And

$$\text{FD}_1 : = \lim_{q\to 1} \text{FD}_q = \exp\left(-\sum_{i=1}^{N} f_i \log f_i\right). \tag{6}$$

Note that the Gini-Simpson index (GSI) used in the main text is related to $\text{FD}_2$ as follows:

$$\text{GSI} : = 1 - \sum_{i=1}^{N} f_i^2 = 1 - \frac{1}{\text{FD}_2}. \tag{7}$$

### Nutritional diversity
Consider a subject' food profile of $N$ foods with the relative abundance given by a vector $\boldsymbol{f} = [f_1, \cdots, f_N]$ and pair-wise nutritional distance matrix $\triangle = (d_{ij}) \in \mathbb{R}^{N\times N}$ with $d_{ii} = 0$ for all $i = 1, \cdots, N$ and $d_{ij} = d_{ji} \geq 0$ for all $i \neq j$. Follow our former definition of functional redundancy, we use the new pair-wise nutritional distance matrix to overcome the drawbacks of original distance matrix

$$\triangle' = \left(d'_{ij}\right) = \begin{pmatrix} \lambda_1 & d_{12} & d_{13} & \cdots & d_{1N} \\ d_{21} & \lambda_2 & d_{23} & \cdots & d_{2N} \\ d_{31} & d_{32} & \lambda_3 & \cdots & d_{3N} \\ \vdots & \vdots & \vdots & \ddots & \vdots \\ d_{N1} & d_{N2} & d_{N3} & \cdots & \lambda_N \end{pmatrix}, \tag{8}$$

where $d_{ij}$ represents the original nutritional distance between food-$i$ and $j$ and $\lambda_i : = \frac{\sum_{j\neq i}^{N} d_{ij}}{N-1}$ is the average nutritional distance between food-$i$ and all other foods. Note that, when different species are equally distinct with a constant pairwise distance, $\lambda$ is equal to this constant. The final nutritional diversity of a subject can be defined as

$$\text{ND}_q(Q') : = D_q(Q') \cdot Q' = \left(\sum_{i=1}^{N}\sum_{j=1}^{N} \frac{d'_{ij}}{Q'}\left(f_i f_j\right)^q\right)^{\frac{1}{2(1-q)}} \cdot Q' \text{ for } q \neq 1, \tag{9}$$

and

$$\mathrm{ND}_1(Q') := D_1(Q') \cdot Q' = \exp\left[-\frac{1}{2}\sum_{i=1}^{N}\sum_{j=1}^{N}\frac{d'_{ij}}{Q'}f_i f_j \log(f_i f_j)\right] \cdot Q'. \quad (10)$$

## Nutritional redundancy

In the literature of ecology, the functional redundancy (FR) of a sample is often considered to be the part of the taxonomic diversity (TD) that cannot be explained by the functional diversity (FD)[48]. Hence FR is typically defined to be the difference between TD and FD. Similarly, we define the NR to be the difference between food diversity (FD) and nutritional diversity (ND):

$$\mathrm{NR} := \mathrm{FD} - \mathrm{ND}. \quad (11)$$

In the main text, we chose FD to be the Gini-Simpson index: $\mathrm{GSI} \equiv 1 - \sum_{i=1}^{N}f_i^2 = \sum_{i=1}^{N}\sum_{j\neq i}^{N}f_i f_j$, and ND was chosen to be the Rao's quadratic entropy $Q \equiv \sum_{i=1}^{N}\sum_{j\neq i}^{N}d_{ij}n_i n_j$. Hence,

$$\mathrm{NR} = \mathrm{GSI} - Q = 1 - \sum_{i=1}^{N}f_i^2 = \sum_{i=1}^{N}\sum_{j\neq i}^{N}(1 - d_{ij})f_i f_j. \quad (12)$$

For the Hill number-based food diversity $\mathrm{TD}_q$ and nutritional diversity $\mathrm{ND}_q(Q')$, and define a parametric class of nutritional redundancy:

$$\mathrm{NR}_q(Q') := \mathrm{TD}_q - \mathrm{ND}_q(Q'). \quad (14)$$

For $q \neq 1$,

$$\mathrm{NR}_q(Q') := \left(\sum_{i=1}^{N}f_i^q\right)^{\frac{1}{(1-q)}} - \left(\sum_{i=1}^{N}\sum_{j=1}^{N}\frac{d'_{ij}}{Q'}(f_i f_j)^q\right)^{\frac{1}{2(1-q)}} \cdot Q'. \quad (15)$$

For $q = 1$,

$$\mathrm{NR}_1(Q') := \exp\left(-\sum_{i=1}^{N}f_i \log f_i\right) - \exp\left[-\frac{1}{2}\sum_{i=1}^{N}\sum_{j=1}^{N}\frac{d'_{ij}}{Q'}f_i f_j \log(f_i f_j)\right] \cdot Q'. \quad (16)$$

## Definition of disease outcomes

Type 2 diabetes was confirmed if at least one of the following criteria by the National Diabetes Data Group[75] was met: (1) elevated plasma glucose levels (fasting glucose ≥**140** mg/dL or random glucose ≥**200** mg/dL) with ≥**1** classic symptoms (polydipsia, polyuria, polyphagia, weight loss, or coma); (2) elevated plasma glucose on at least two occasions (fasting glucose ≥**140** mg/dL, random glucose ≥**200** mg/dL, and/or glucose ≥**200** mg/dL after an oral glucose test) with no symptoms; and (3) hypoglycemic therapy with insulin or oral medications. An updated cutoff of 126 mg/dL for fasting glucose was used for diagnoses after 1997 according to the American Diabetes Association diagnostic criteria[76].

The primary outcome measure was major Cardiovascular disease (CVD), which is defined as a combined endpoint of non-fatal myocardial infarction, non-fatal stroke, or fatal CVD (fatal stroke, fatal myocardial infarction, and other cardiovascular death). The secondary outcome measures were assessed as following: (1) total CHD: defined as fatal CHD and nonfatal myocardial infarction; (2) total stroke: all fatal and nonfatal stroke cases (ischemic, hemorrhagic, and undetermined subtypes). When a participant (or family

members of deceased participants) reported an incident event, permission was requested to examine their medical records by physicians who were blinded to the participant risk factor status. For each endpoint, the month and year of diagnosis were recorded as the diagnosis date. Non-fatal events were confirmed through review of medical records. Myocardial infarction was confirmed if the World Health Organization criteria were met on the basis of symptoms plus diagnostic electrocardiogram changes or elevated cardiac enzymes. If medical records were unavailable, we considered myocardial infarctions probable when the participant provided additional confirmatory information. Information on angina and coronary revascularization procedures (percutaneous transluminal coronary angioplasty or coronary artery bypass grafting surgery) were self-reported, and we included only events that occurred before a manifest cardiovascular event.

Strokes were confirmed if data in the medical records fulfilled the National Survey of Stroke criteria requiring evidence of a neurological deficit with sudden or rapid onset that persisted for >24 h of until death[77]. We excluded cerebrovascular pathology due to infection, trauma, or malignancy, as well as "silent" strokes discovered only by radiologic imaging. Radiology reports of brain imaging (computed tomography or magnetic resonance imaging) were available in 89% of those with medical records. We classified strokes as ischemic stroke (thrombotic or embolic occlusion of a cerebral artery), hemorrhagic stroke (subarachnoid and intraparenchymal hemorrhage), or stroke of probable/unknown subtype (a stroke was documented but the subtype could not be ascertained owing to medical records being unobtainable).

Deaths were identified by reports of families, the U.S. postal authorities, and searches of the National Death Index. The cause of death was assigned by physicians after review of medical records and death certificate information. Follow-up for deaths was >98% complete. Fatal CVD was defined as fatal CHD disease, fatal stroke, or fatal CVD. Fatal CHD was defined as ICD-9 (international classification of diseases, ninth revision) codes 410-412 and was considered confirmed if fatal CHD was confirmed via medical records or autopsy reports or if CHD was listed as the cause of death on the death certificate and there was prior evidence of CHD in the medical records. We designated as probable those cases in which CHD was the underlying cause on the death certificates, but no prior knowledge of CHD was indicated and medical records concerning the death were unavailable. Similarly, we used ICD-9 codes 430–434 to define fatal stroke and followed the same procedures to classify cases of confirmed or probable fatal stroke. Lastly, fatal CVD was defined by ICD-9 codes 390–458.

## Reporting summary

Further information on research design is available in the Nature Portfolio Reporting Summary linked to this article.

## Data availability

The DMAS dataset is available from https://github.com/knights-lab/dietstudy_analyses. The USDA Food Nutrient Database is available from https://fdc.nal.usda.gov/download-datasets.html. The Women's lifestyle validation study, Men's lifestyle validation study, Nurses' Health Study (NHS) and Health Professionals Follow-Up Study (HPFS) are not publicly available for the following reason: data contain information that could compromise research participant privacy. Requests to access these data (for research purposes only) can be made by research investigators 1 year after publication via http://www.nurseshealthstudy.org/researchers. Investigators can expect initial responses within 4 weeks of request submission. Harvard University Food Composition Database can be accessed at https://regepi.bwh.harvard.edu/health/nutrition.html. FoodDB database can be accessed

at https://foodb.ca/downloads. Source data supporting all our findings are provided with this publication as a Source Data file. Source data are provided in this paper. Source data are provided with this paper.

## Code availability

MATLAB Code (version R2016b) used in this work is available at https://github.com/spxuw/Nutritional-redundancy or under Zenodo at https://doi.org/10.5281/zenodo.7781521[78].

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

## Acknowledgements

We thank Laura Sampson, Walter C. Willett, Rui Song, and Jorge Chavarro for valuable discussions. Research reported in this publication was supported by grants UM1CA186107, U01CA167552, K25HL166208, R01AI141529, R01HD093761, RF1AG067744, UH3OD023268, U19AI095219, and U01HL089856 from National Institutes of Health.

## Author contributions

Y.-Y.L. conceived and designed the project. X.-W.W. performed all the numerical calculations and real data analyses. X.-W.W. and Y.H. performed the disease association analyses. X.-W.W. and Y.-Y.L. analyzed and interpreted the results with the assistance from G.M., F.G., S.N.B., Q.S., X.Z., F.B.H., and S.T.W. X.-W.W. and Y.-Y.L. wrote the manuscript. All authors edited the manuscript.

## Competing interests

The authors declare no competing interests.

### Ethical approval

The NHS and HPFS study protocols were approved by the institutional review board (IRB) of the Brigham and Women's Hospital, and the IRB allowed participants' completion of questionnaires to be considered as implied consent. Written informed consent was obtained from participants to release medical records documenting. The WLVS and MLVS studies were approved by the human subjects committees of the Harvard T.H. Chan School of Public Health and Brigham and Women's Hospital (Boston, Massachusetts). The DMAS study was approved by the University of Minnesota Institutional Review Board.

### Informed consent

Informed consent was obtained from all study subjects prior to any study procedures.
