## [Peer Review File · Nature Communications]

Reviewer comments, first round -

Reviewer #1 (Remarks to the Author):

This paper entitled "Nutritional Redundancy in Human Diet" by Wang et al. represents development and evaluation of a novel concept referred to as nutritional redundancy in nutrition research. The concept is novel and links variation in food intake with variability in nutrient intake at an individual's level and its relevance for prediction of health outcomes. It has many similarities with the concept of functional redundancy in microbiome research. The authors constructed food-nutrient network, i.e. a bipartite graph connecting foods to the nutrients they contain and used network science tools to elucidate topological features that contribute to the nutrient redundancy in the diet of humans. A reference food nutrient network was constructed to visualize the topological features of a real food nutrient network using USDA's Food and Nutrition Database for Dietary Studies 2011-212, consisting of 7618 foods and 65 nutrients and micronutrients. They also used the food-nutrient network to calculate and define individual's nutritional redundancy and found that this measure (nutritional redundancy) was not strongly linked to commonly used healthy eating scores. To demonstrate the nutritional redundancy phenomenon, the authors used dietary data and assessments from 4 different studies representing different time perspectives (from days to years) and different dietary assessment methods (self-administered 24h dietary assessment tool and FFQs). The authors also evaluated the performance of nutritional redundancy and the selected healthy eating scores in prediction of healthy gaining and risk of developing CVD and T2D. They found that nutritional redundancy, which represents a measure of the difference between food diversity and nutrient diversity of a person's dietary assessment, similarly associated with healthy ageing as the commonly used diet index scored compared. The authors conclude that personal nutritional redundancy could serve as a powerful tool in nutrition science. Several potential limitations were mentioned but not further investigated.

To my opinion, this paper provides a novel concept which is well described, extensively evaluated across different time scales and different dietary assessment methods/databases and which was also tested in relation to meaningful outcomes in comparison with commonly used diet scores. Other groups also work on food diversity scores and have tried similar approaches but not yet developed in such a nice and elegant way as presented here. The paper is well written, and figures and tables are overall fine. Supplemental material was also helpful to decipher the to complete picture. However, although the concept of nutritional redundancy represents a novel development and could be a useful and interesting tool for risk modelling in nutritional epidemiological research, it is, at this stage, difficult to see how it can be translated into practical implications in nutrition and public health. This is an aspect which was not discussed in detail by the authors. It would be good to add such discussion to the manuscript, i.e., how the concept of nutritional redundancy can be translated into advice or at least testable nutritional concepts at intervention level. For natural reasons, development of such concrete advice is too early, but a general discussion about the translation of the NR-concept and findings thereof into practical nutritional guidance or testable interventions could preferably be added to the manuscript.

Since NR and healthy diet scores appears to be orthogonal, would it be possible to combine the concepts to better capture the total impact of diet on health outcomes? If so, any ideas of how such combined measure could look like? This could also be elaborated on in the discussion.

Specific comments:

The title could perhaps indicate more clearly that Nutritional Redundancy is introduced as a new concept and perhaps even for what purpose. As it is now it may not capture the deserved attention.

Line 28: I would add in a "may" offers. Since this has not be thoroughly tested across studies.

Line 65: "are still be met". Is any word missing here?

Line 81: these are results. Move to results section?

Line 215-219: What is the implication of normalizing NR in the way conducted? Did you test to what degree food diversity affected NR? Intuitively, a high food diversity will contribute high NR. If

comparing NR across widely different populations with large differences in diet diversities, how can the NR be interpreted if removing effects of FD? Isn't food diversity an inherent feature of NR, i.e. NR is conditioned on FD?

Line 265-266: "suggests that the human diet prefers low nutritional redundancy". Can a human diet prefer anything? Pls reformulate.

Line 302-318: How were these prediction models validated (beyond comparison with healthy food indices)? Why not replicating in HPFS for example?

Reviewer #2 (Remarks to the Author):

In this study, the authors introduced the concept of NR and investigated its relation with FNN and individual health status like diabetes. Although the results sound interesting, I raise the following comments about the rigor of their approach and result interpretation:

1. I believe that additional efforts should be given to more sound and rigorous explanations and understanding of the major findings in this study, to "demystify" these results. It can be done by fair and rigorous examination of alternative explanations of those results, rather than staying in the single possibilities claimed in this study.

For example, it should be more elucidated why NR serves as a good indicator of lower diabetes incidence, because this study does not seem to provide its explanation. In fact, one can suspect that this may be a "byproduct" of other factors, rather than by NR itself. For a person with high NR, his/her consumed foods may be "densely" filled in the food network based on the foods' nutritional similarity, and therefore high NR can be a mere byproduct of a large diversity of foods enjoyed by this individual. Or, it may be the byproduct of his/her tendency to use unprocessed food items with their variable combinations in the diet rather than to rely on the constant processed food packages or restaurant foods that may have much less diverse or variable food items. To me, the association of such diverse food repertoire (or the use of unprocessed foods cooked at home) in one's diet with the low diabetes risk makes more sense based on my existing knowledge, than the NR's itself which is not clear from this paper. Although the authors wrote that they used normalized NR (normalized by food diversity), i.e. nNR, but it itself does not exclude the possibility that nNR still positively correlates with the diversity of the consumed food items as I supposed above. I wonder whether the authors measured the correlation between nNR and the diversity of the consumed food items, and if they are positively correlated, the latter will likely explain the low diabetes risk than the nNR itself. Similar analysis can be conducted for the consumption pattern of unprocessed foods and diabetes risk, as well. My point is that, if the authors believe that NR is a real indicator of low diabetes risk, they need to provide the sound reasoning for this; otherwise, the possibilities of other factors are still open, as I gave their examples here.

In a similar line, I also question the relation between high NR and the nestedness of FNN. Did the authors examine whether there is a simpler reason than the nestedness, to explain high NR? Although they examined the null models like randomized networks with conserved degrees, I suppose that there should be more tangible reason for high NR without the need to jump into the concept of nestedness. A food network based on the foods' nutritional similarity is already known to be highly modular, separately clustered around animal-based foods and plant-based foods at first, and fish and meats are separately clustered in animal-based food cluster and grains, fruits, vegetables, nuts are separately clustered in plant-based food cluster, etc (PLOS ONE 10, e0118697 (2015)). Isn't such clustered nature itself in the network (rather than nestedness) close to the major reason for high NR, as individual people are likely to have their preference of the ratio of (let's say) protein-rich foods to carbohydrate-rich foods, which roughly match the individual modules (clusters) in the food network? (PLOS ONE 10, e0118697 (2015)) I suppose that it may be analyzable by another null model by preserving the network clusters while destroying their nestedness.

In addition, I wonder the origin of the nestedness in FNN itself. If the authors describe specific examples of some food items and their nutrient contents, it may be helpful to understand their nestedness in the network, though. One possibility may be that cooked foods have trivially more food ingredients inside, so they can have more nutrients as a superset of the nutrients of other

raw foods. Also, even among the uncooked foods, there are a number of foods that trivially share very similar nutritional compositions, like the relation between raw food, its dried food, the frozen food, salted food, fortified food, etc. I wonder whether the consolidation of these trivially redundant foods, and focusing on only raw foods without cooked foods, will still reproduce the nestedness in the network.

2. I have the other following concerns about the rigor of the methods in this study.

(1) When the authors define nutritional distance between foods, they used weighted Jaccard distance, including the sum of min (...) divided by the sum of max (...). However, in this formula, both the numerator and denominator are simply likely to be dominated by particular nutrients with typically large amounts in food, because this formula is based on the sheer summation of these amounts and the nutrient amounts vary a lot across different nutrients. The summation in the numerator or denominator may be just a reflection of macronutrient amounts, with relatively little contribution of low-amount nutrients like vitamins and minerals. Because different nutrients have their own scales of the amounts and these scales vary vastly across different nutrients, I suggest the authors to redefine the distance between foods, considering these nutrient-specific scales.

(2) It is unclear to me how the authors avoided the redundancy issues among nutrients in their analysis. For example, to my experience, the concentrations of individual amino acids are highly positively correlated with each other across different foods. Therefore, considering all these 20 amino acids may "over-represent" their effects, and thus considering only the total protein amount may suffice for this study. In a similar sense, some food database provides the information of many different types of lipids (different in their carbon numbers, etc) in food, and I wonder how the authors treated such problems that can introduce systematic biases into a particular group of nutrients with many redundant subtypes.

(3) In this study, the authors seem to have defined the relative abundance of each nutrient by the normalization of the "summation" of these nutrient amounts inside the food (L115). However, such normalization factor can possibly depend on the particular choice of considered nutrients and susceptible to the points in (2) above. Better normalization factor may be dry food weight (total food weight – water weight inside that food), to make relative abundance of the nutrient as nutrient weight per dry food weight. Alternatively, one may try total food weight as a normalization factor rather than dry food weight, but it may introduce spurious nutritional correlations between different foods solely based on their water contents (PLOS ONE 10, e0118697 (2015)).

(4) In this paper, the word "personalized" is often used (e.g., L116 and other places). However, I think that this word does not seem to be rigorously defined in this study. My sense of "personalization" is that intra-individual similarity of the foods consumed over time should be larger than inter-individual similarity of their consumed foods, but I wonder whether the authors used any measures to really quantify such tendency.

(5) Overall, I believe that the authors need to provide more specific examples of their findings, to make the readers more convinced. For example, in L119-120 "The most abundant food items of individuals in most studies were from the sweetened beverages, and vegetables." Are the vegetables really one of the most abundant food items in their diet data? (the subjects in these cohorts may mainly have practiced so-called Western diets, though?)

(6) The authors performed some statistical tests to examine the significance of their results, but sometimes they took t-test, sometimes U-test, without the explanation of the choices of these methods. For example, is there any normality in their data to justify their use of t-test or z-test? If yes, why did they then use U-test (free of normality condition) sometimes?

(7) It is still unclear to me "how" the authors controlled for possible confounding effects in their analysis, although they mentioned that they controlled for them.

3. In L146, I suppose that n_i and n_j in the Rao's quadratic entropy Q shall be corrected as f_i and f_j . Also, the font sizes in Fig. 3 need to be increased for better visual inspection.

Response to Reviewer 1

Point 1.0: This paper entitled "Nutritional Redundancy in Human Diet" by Wang et al. represents development and evaluation of a novel concept referred to as nutritional redundancy in nutrition research. The concept is novel and links variation in food intake with variability in nutrient intake at an individual's level and its relevance for prediction of health outcomes. It has many similarities with the concept of functional redundancy in microbiome research. The authors constructed food-nutrient network, i.e. a bipartite graph connecting foods to the nutrients they contain and used network science tools to elucidate topological features that contribute to the nutrient redundancy in the diet of humans. A reference food nutrient network was constructed to visualize the topological features of a real food nutrient network using USDA's Food and Nutrition Database for Dietary Studies 2011-2012, consisting of 7618 foods and 65 nutrients and micronutrients. They also used the food-nutrient network to calculate and define individual's nutritional redundancy and found that this measure (nutritional redundancy) was not strongly linked to commonly used healthy eating scores. To demonstrate the nutritional redundancy phenomenon, the authors used dietary data and assessments from 4 different studies representing different time perspectives (from days to years) and different dietary assessment methods (self-administered 24h dietary assessment tool and FFQs). The authors also evaluated the performance of nutritional redundancy and the selected healthy eating scores in prediction of healthy ageing and risk of developing CVD and T2D. They found that nutritional redundancy, which represents a measure of the difference between food diversity and nutrient diversity of a person's dietary assessment, similarly associated with healthy ageing as the commonly used diet index score compared. The authors conclude that personal nutritional redundancy could serve as a powerful tool in nutrition science. Several potential limitations were mentioned but not further investigated.

Response: We thank Reviewer #1 for reviewing our manuscript, and her/his very positive assessment on the novelty of our work. Next, we address each of the reviewer's comments in order.

Point 1.1: To my opinion, this paper provides a novel concept which is well described, extensively evaluated across different time scales and different dietary assessment methods/databases and which was also tested in relation to meaningful outcomes in comparison with commonly used diet scores. Other groups also work on food diversity scores and have tried similar approaches but not yet developed in such a nice and elegant way as presented here. The paper is well written, and figures and tables are overall fine. Supplemental material was also helpful to decipher the complete picture. However, although the concept of nutritional redundancy represents a novel development and could be a useful and interesting tool for risk modelling in nutritional epidemiological research, it is, at this stage, difficult to see how it can be translated into practical implications in nutrition and public health. This is an aspect which was not discussed in detail by the authors. It would be good to add such discussion to the manuscript, i.e., how the concept of nutritional redundancy can be translated into advice or at least testable nutritional concepts at intervention level. For natural reasons, development of such concrete advice is too early, but a general discussion about the translation of the NR-concept and findings thereof into practical nutritional guidance or testable interventions could preferably be added to the manuscript.

Response: We thank Reviewer #1 for this very constructive comment. In the revised manuscript, we have added the following discussion about the potential translation of the NR-concept and findings into practical nutritional guidance (see main text, page 12, lines 400-404):

“Moreover, we examined its associations with the risks of type 2 diabetes and cardiovascular disease in NHS (all female) and HPFS (all male). For both cohorts, we found a clear inverse association between NR and the two phenotypes after adjusting for age. For HPFS, the inverse association is observed even after adjusting for a wide range of confounding factors. Whether these findings can lead to practical nutritional guidance warrant further interventional studies.”

Point 1.2: Since NR and healthy diet scores appears to be orthogonal, would it be possible to combine the concepts to better capture the total impact of diet on health outcomes? If so, any ideas of how such combined measure could look like? This could also be elaborated on in the discussion.

Response: We thank Reviewer #1 for this very constructive comment. Indeed, we can combine the concepts of NR with existing diet scores. In the revised manuscript, we added the following discussion (see main text, page 12, lines 405-411):

“Since the personal NR measure is not strongly correlated with any classical healthy diet scores, in principle we can combine the concepts of NR and those healthy diet scores to better capture the total impact of diet on health outcomes. For instance, one can leverage the food-specific subgraphs of the FNN (see **Fig.S2**) to calculate the NR of food groups contributing to each component of a healthy diet score. This will enable us to define an NR-aware healthy diet score. Systematically exploring this direction warrant dedicated efforts, which is beyond the scope of the current work.”

Point 1.3:

Specific comments:

The title could perhaps indicate more clearly that Nutritional Redundancy is introduced as a new concept and perhaps even for what purpose. As it is now it may not capture the deserved attention.

Response: We thank Reviewer #1 for this very constructive comment. We have revised the title as: “Nutritional redundancy in the human diet and its application in phenotype association studies”.

Point 1.4: Line 28: I would add in a “may” offers. Since this has not be thoroughly tested across studies.

Response: We thank Reviewer #1 for this suggestion. We have revised that sentence accordingly (see main text, page 1, line 33).

Point 1.5: Line 65: “are still be met”. Is any word missing here?

Response: Sorry for the typo. In the revised manuscript (see main text, page 2, lines 65-67), we have revised that sentence as

“Although dietary patterns sometimes exhibit drastically divergent food choices, at a finer resolution, the underlying nutrient palettes could reveal a higher degree of similarity.”

Point 1.6: Line 81: these are results. Move to results section?

Response: We thank Reviewer #1 for this comment. We have simplified that paragraph into one sentence (see main text, page 3, lines 75-76):

“In this work, we aim to reveal the origin of NR in the human diet and explore its power in phenotype association studies.”

Point 1.7: Line 215-219: What is the implication of normalizing NR in the way conducted? Did you test to what degree food diversity affected NR? Intuitively, a high food diversity will contribute high NR. If comparing NR across widely different populations with large differences in diet diversities, how can the NR be interpreted if removing effects of FD? Isn't food diversity an inherent feature of NR, ie NR is conditioned on FD?

Response: We thank Reviewer #1 for this critical comment.

In this work, we define the NR of a dietary assessment from a particular individual as the part of its food diversity (FD) that cannot be explained by its nutrient diversity (ND), i.e., $NR = FD - ND$. So, by definition, NR is conditioned on FD. To simplify the result presentation and avoid confusion, in the revised manuscript, we have replaced nNR by NR in all the calculations, finding that our conclusions still hold (see **Figs.R1-R2** and **Tables R1-R2**, corresponding to Figure 4, Figure 6, and Tables 1-2 in the revised manuscript).

Point 1.7: Line 265-266: “suggests that the human diet prefers low nutritional redundancy”. Can a human diet prefer anything? Pls reformulate.

Response: We thank Reviewer #1 for this comment. In the revised manuscript, we have removed that sentence.

Point 1.8: Line 302-318: How were these prediction models validated (beyond comparison with healthy food indices)? Why not replicating in HPFS for example?

Response: We thank Reviewer #1 for this very constructive comment. In the previous manuscript, the

prediction model of healthy aging was validated by a standard cross-validation. We used 80% participants to train the classifier and the remaining 20% participants to validate the model.

Following the reviewer's suggestion, in the revised manuscript we also replicated the prediction models in HPFS. The healthy aging in HPFS was defined in a similar way [1] as in NHS: (1) no self-reported history of cancer, diabetes, CHD, coronary artery bypass graft surgery or percutaneous trans-luminal coronary angioplasty, congestive heart failure, stroke, kidney failure, chronic obstructive pulmonary disease, Parkinson disease, multiple sclerosis, or amyotrophic lateral sclerosis; (2) no cognitive decline (subjective cognitive decline score = 0 by 6 relevant questions); and (3) no physical limitations (no limitations on moderate activities, and no more than moderate limitations on more demanding physical performance measures of the 36-item Medical Outcomes Study Short- Form Health Survey).

We used NR (or one of the other four healthy eating indices) and related covariates to predict the healthy agers versus normal agers. As shown in **Fig.R3**, the result is similar as what we found in NHS.

In the revised manuscript, we have added the above result (see main text, page 10, lines 335-339):

“We also performed the healthy aging prediction using data from a substudy of HPFS with 6,160 healthy agers and 11,534 usual agers⁶⁴. Again, we used personal NR or one of the four healthy diet scores in 1998 and other host factors to predict the healthy aging status. We found that NR can also achieve very similar error rate (or AUROC) as other healthy diet scores in HPFS. Moreover, the performance of NR in HPFS is comparable to that in NHS (**Fig.S10**).”

Inspired by the reviewer's comment, we also replicated the association studies on NR and the risk of cardiovascular disease and type 2 diabetes in HPFS, finding similar results as observed in NHS. In the revised manuscript, we have added these new results (see main text, page 11, lines 365-369):

“For HPFS participants, we observed similar results. For type 2 diabetes, after adjusting for age (months) only, the P for trend < 0.001; after adjusting for a wide range of confounding factors, the P for trend = 0.002 (see **Table 1**). For cardiovascular disease, after adjusting for age (months) only, the P for trend = 0.004; after adjusting for a wide range of confounding factors, the P for trend = 0.04 (see **Table 2**).”

Finally, we thank Reviewer #1 again for reviewing our manuscript and her/his very insightful and constructive comments, which help us greatly improve our work. We hope our responses have addressed her/his comments in a satisfactory manner.

Response to Reviewer 2

Point 2.0: In this study, the authors introduced the concept of NR and investigated its relation with FNN and individual health status like diabetes. Although the results sound interesting, I raise the following comments about the rigor of their approach and result interpretation:

Response: We thank Reviewer #2 for reviewing our manuscript and finding our results interesting. Next, we address each of her/his comments in order.

Point 2.1: 1. I believe that additional efforts should be given to more sound and rigorous explanations and understanding of the major findings in this study, to “demystify” these results. It can be done by fair and rigorous examination of alternative explanations of those results, rather than staying in the single possibilities claimed in this study.

For example, it should be more elucidated why NR serves as a good indicator of lower diabetes incidence, because this study does not seem to provide its explanation. In fact, one can suspect that this may be a “byproduct” of other factors, rather than by NR itself. For a person with high NR, his/her consumed foods may be “densely” filled in the food network based on the foods’ nutritional similarity, and therefore high NR can be a mere byproduct of a large diversity of foods enjoyed by this individual. Or, it may be the byproduct of his/her tendency to use unprocessed food items with their variable combinations in the diet rather than to rely on the constant processed food packages or restaurant foods that may have much less diverse or variable food items. To me, the association of such diverse food repertoire (or the use of unprocessed foods cooked at home) in one’s diet with the low diabetes risk makes more sense based on my existing knowledge, than the NR’s itself which is not clear from this paper. Although the authors wrote that they used normalized NR (normalized by food diversity), i.e. nNR, but it itself does not exclude the possibility that nNR still positively correlates with the diversity of the consumed food items as I supposed above. I wonder whether the authors measured the correlation between nNR and the diversity of the consumed food items, and if they are positively correlated, the latter will likely explain the low diabetes risk than the nNR itself. Similar analysis can be conducted for the consumption pattern of unprocessed foods and diabetes risk, as well. My point is that, if the authors believe that NR is a real indicator of low diabetes risk, they need to provide the sound reasoning for this; otherwise, the possibilities of other factors are still open, as I gave their examples here.

Response: We thank Reviewer #2 for this very insightful and constructive comment.

Following the reviewer’s excellent suggestion, we calculated the Spearman correlation coefficient ρ between nutritional redundancy (NR) (or the normalized NR, i.e., $nNR=NR/FD$) and food diversity (FD). As shown in **Fig.R4**, for all the five studies (DMAS, WLVS, MLVS, NHS and HPFS), both NR and nNR are positively correlated with FD, and the correlation between nNR and FD is weaker than the correlation between NR and FD. This result suggests that, for those studies, high NR is indeed partially due to high FD and this correlation is partially eliminated by normalizing NR with FD.

To check if FD itself can serve as an indicator of low risk of type 2 diabetes and cardiovascular disease, we performed the association studies of FD and the risk of the two diseases in NHS and HPFS, just as we did for the NR-disease association analysis. Interestingly, for both NHS and HPFS participants, we found that, after adjusting for a wide range of confounding factors, FD is not associated with lower risk of type 2 diabetes or cardiovascular disease at all (see **Tables R3-R4**). This result implies that the association between NR and disease risks cannot be simply attributed to FD.

To understand the association between NR and the risk of type 2 diabetes and cardiovascular disease in the two cohorts, we analyzed the food consumption pattern of their participants. In particular, we divided the NHS (and HPFS) participants into tertiles (denoted as T1, T2, T3) based on their NR values (T3 has the highest NR). For both NHS and HPFS participants, we found that there is a consistent trend among the three NR tertiles for those important food groups (which are involved in the computation of traditional healthy eating indices) (see **Fig.R5**). For instance, the abundances of Fruits, Vegetables, Dairy, and Cereal Grains are much higher for T3 (i.e., high-NR participants) than T2 and T1; while the abundances of Beverages are much lower for T3 than T2 and T1. These results might explain why NR can serve as a good indicator of lower risk for type 2 diabetes and cardiovascular disease for NHS and HPFS participants.

In the revised manuscript, we have added the above results (see main text, page 11, lines 373-386):

“Since NR is a part of FD and actually they are positively correlated (see Fig.S11), we wonder if FD itself is associated with the risk of type 2 diabetes and cardiovascular disease. We performed association analyses. Interestingly, for both NHS and HPFS participants, we found that, after adjusting for a wide range of confounding factors, FD is not associated with lower risk of type 2 diabetes (see **Table S4**) or cardiovascular disease (see **Table S5**) at all. This result implies that the association between NR and disease risks cannot be simply attributed to FD.

To understand the association between NR and the risk of type 2 diabetes and cardiovascular disease in the two cohorts, we analyzed the food consumption pattern of each NR tertiles in those two cohorts. We found that there is a consistent trend among the three NR tertiles for those important food groups in NHS and HPFS (see **Fig.7**). For instance, abundances of Fruits, Vegetables, Dairy, Cereal Grains are much higher for T3 (i.e., high-NR participants) than T2 and T1; while abundances of Beverages are much lower for T3 than T2 and T1 in both NHS and HPFS. This food consumption pattern might explain why NR is an indicator of low risk of type 2 diabetes and cardiovascular disease.”

Point 2.2: In a similar line, I also question the relation between high NR and the nestedness of FNN. Did the authors examine whether there is a simpler reason than the nestedness, to explain high NR? Although they examined the null models like randomized networks with conserved degrees, I suppose

that there should be more tangible reason for high NR without the need to jump into the concept of nestedness. A food network based on the foods' nutritional similarity is already known to be highly modular, separately clustered around animal-based foods and plant-based foods at first, and fish and meats are separately clustered in animal-based food cluster and grains, fruits, vegetables, nuts are separately clustered in plant-based food cluster, etc (PLOS ONE 10, e0118697 (2015)). Isn't such clustered nature itself in the network (rather than nestedness) close to the major reason for high NR, as individual people are likely to have their preference of the ratio of (let's say) protein-rich foods to carbohydrate-rich foods, which roughly match the individual modules (clusters) in the food network? (PLOS ONE 10, e0118697 (2015))

Response: We thank Reviewer #2 for this critical comment, and we apologize for not citing this very important reference. As we shown in the Figure 5 of main text, we constructed a food similarity network based on the nutrient context in each food. We do see clear clusters contributed by each food group, which is consistent with what was reported in (PLOS ONE 10, e0118697 (2015)).

In the revised manuscript, we have cited this important reference and added the following sentences (see main text, page 7, lines 238-243):

“We found a clear modular structure in the food similarity network, i.e., food items from the same food group form a densely connected cluster or module (see **Fig.5a**), which is consistent with previous study that a food network based on the foods' nutritional similarity displays separately clustered around animal-based foods and plant-based foods at first, and fish and meats are separately clustered in animal-based food cluster and grains, fruits, vegetables, nuts are separately clustered in plant-based food cluster⁵⁴.”

There are several reasons why we chose to **directly** study the topological features of the bipartite food-nutrient network (FNN). First, the unipartite projection of the FNN is not unique. Indeed, depending on the similarity or edge threshold, one can construct a series of unipartite graphs, i.e., the food similarity network (FSN). Second, the topological features of the bipartite graphs can be studied without any ambiguities. In particular, the nested structure is a commonly observed property of bipartite graphs (e.g., plant-pollinator network in ecology^{2,3}, gene content network in microbiology⁴). Therefore, it would be very natural to study the impact of nested structure on NR.

Regarding whether there is a simpler reason than the nestedness to explain high NR, we did perform modularity analysis of the bipartite FNN using the Beckett method⁷. We found that the modularity Q of the FNN is 0.11, which is much lower than that of other bipartite graphs (e.g., plant-pollinator networks typically with $Q > 0.5$)⁷. In other words, the modular structure of the FNN is weak, and we don't think it can explain the high NR.

Point 2.3: I suppose that it may be analyzable by another null model by preserving the network clusters while destroying their nestedness.

Response: We thank Reviewer #2 for this comment. As explained in our response to Point 2.2, in this study, we are directly considering the topological features of the bipartite FNN, rather than its unipartite projections. We have tried 4 basic null models of the bipartite graph. Systematically exploring other null models of the bipartite graph (or its unipartite presentations) deserves dedicated effort, which is beyond the scope of the current study.

Point 2.4: In addition, I wonder the origin of the nestedness in FNN itself. If the authors describe specific examples of some food items and their nutrient contents, it may be helpful to understand their nestedness in the network, though. One possibility may be that cooked foods have trivially more food ingredients inside, so they can have more nutrients as a superset of the nutrients of other raw foods. Also, even among the uncooked foods, there are a number of foods that trivially share very similar nutritional compositions, like the relation between raw food, its dried food, the frozen food, salted food, fortified food, etc. I wonder whether the consolidation of these trivially redundant foods, and focusing on only raw foods without cooked foods, will still reproduce the nestedness in the network.

Response: We thank Reviewer #2 for this insightful comment. We fully agree with Reviewer #2 that those cooked foods have more food gradients than that in raw foods, as the ingredients for those cooked/recipes are calculated from the sum weights of ingredients of recipe with considering the retention factor⁵.

In addition, we analyzed the FNN of 51 raw foods in Harvard food nutrient database and we found that the network still displays high nestedness structure. However, NODF of this raw FNN is 0.573 (see **Fig.R6**), which is lower than that of the original FNN (higher than 0.8). In Fig.R6, we also highlighted two foods (1) raw celery, (2) raw onions, spring or scallions, which share very similar nutritional compositions.

In the revised manuscript, we have mentioned this analysis (see main text, page 6, lines 207-209):

“In addition, we analyzed the FNN of 51 raw foods in HFDB and found that this network still displays high nestedness structure. However, NODF of this raw-food FNN is 0.573, much lower than that of the whole FNN (see **Fig.S4**).”

Point 2.5:

2. I have the other following concerns about the rigor of the methods in this study.

(1) When the authors define nutritional distance between foods, they used weighted Jaccard distance, including the sum of $\min(\dots)$ divided by the sum of $\max(\dots)$. However, in this formula, both the numerator and denominator are simply likely to be dominated by particular nutrients with typically large amounts in food, because this formula is based on the sheer summation of these amounts and the nutrient amounts vary a lot across different nutrients. The summation in the numerator or denominator may be just a reflection of macronutrient amounts, with relatively little contribution of low-amount nutrients like vitamins and minerals. Because different nutrients have their own scales of the amounts

and these scales vary vastly across different nutrients, I suggest the authors to redefine the distance between foods, considering these nutrient-specific scales.

Response: We thank Reviewer #2 for this critical comment.

We apologize for the confusion. In the calculation of NR, actually we always used the (**unweighted**) Jaccard distance to measure the nutritional distance between foods, which only concerns if a nutrient is present in a food or not. This way the NR calculation will not be influenced by the nutrient amount variability, especially those high-amount nutrients.

In the previous version of our manuscript, the **weighted** Jaccard index was used only in Figure 5 to project the bipartite food-nutrient network into the unipartite food similarity network. To avoid confusion, in the revised manuscript, we have also used the **unweighted** Jaccard index to perform the projection. So, throughout the revised manuscript, we used **unweighted** Jaccard index (or distance) to measure the nutritional similarity (or distance) between food items.

In the revised manuscript, we added the following sentence (see main text, page 5, lines 153-156):

“For the sake of simplicity, to avoid major effects driven by the several orders of magnitude covered by nutrient amount in food⁴⁹, we compute d_{ij} as the (unweighted) Jaccard distance between the sets of nutrients within two food items (see **SI Sec.2.3** for definition).”

Point 2.6: (2) It is unclear to me how the authors avoided the redundancy issues among nutrients in their analysis. For example, to my experience, the concentrations of individual amino acids are highly positively correlated with each other across different foods. Therefore, considering all these 20 amino acids may “over-represent” their effects, and thus considering only the total protein amount may suffice for this study. In a similar sense, some food database provides the information of many different types of lipids (different in their carbon numbers, etc) in food, and I wonder how the authors treated such problems that can introduce systematic biases into a particular group of nutrients with many redundant subtypes.

Response: We thank Reviewer #2 for this critical comment.

The reason why we did not remove those redundant subtypes in the nutrient group is simply because we wanted to use the original food nutrition database without any pre-processing.

To directly address this nutrient ontology issue, we made two attempts.

- *First*, we demonstrated that the FNN still shows highly nested structure after excluding those nutrients that are not specific enough to have a SIMLES or InChIKey ID, e.g., sugar, total fat, protein, total fiber, etc. As shown in **Fig.R7**, the “perturbed” FNN still demonstrates a highly nested

structure (with nestedness NODF~0.89), which is very close to the nestedness of the original FNN (with NODF=0.91). This result indicates that the nutrient ontology issue doesn't significantly alter the key property (i.e., the nested structure) of the FNN.

- *Second*, we examined whether the nutrient ontology issue affects the NR calculation. For both the USDA and Harvard food nutrition databases, we removed fatty acids-related columns (i.e., total saturated, monounsaturated and polyunsaturated fatty acids) but kept total fat in the food nutrition matrix G . Interestingly, we found that removing fatty acids-related nutrient subtypes did not drastically affect NR: the Pearson correlation between the original NR and the NR after removing fatty acids is 0.99 with p-value < 0.0001 (see **Fig.R8**).

We admit that systematically resolving the nutrient ontology issue in the NR calculation warrants dedicated efforts, which is beyond the scope of our current study. In the revised manuscript, we added the following discussion regarding this point (see main text, page 13, lines 433-438):

“Finally, food distance was measured using all nutrient components, which might introduce biases as some nutrients are correlated or redundant, such as fatty acids and amino acids. Although we have demonstrated that the FNN still shows a highly nested structure after excluding broad families of chemicals (Fig.S5), and that removing fat-related nutrient subtypes does not drastically alter NR (Fig.S12), more efforts are warranted to systematically address the potential impact of nutrition ontology on NR calculation.”

Point 2.7: (3) In this study, the authors seem to have defined the relative abundance of each nutrient by the normalization of the “summation” of these nutrient amounts inside the food (L115). However, such normalization factor can possibly depend on the particular choice of considered nutrients and susceptible to the points in (2) above. Better normalization factor may be dry food weight (total food weight – water weight inside that food), to make relative abundance of the nutrient as nutrient weight per dry food weight. Alternatively, one may try total food weight as a normalization factor rather than dry food weight, but it may introduce spurious nutritional correlations between different foods solely based on their water contents (PLOS ONE 10, e0118697 (2015)).

Response: We thank Reviewer #2 for this very critical comment. The reason why we normalized the nutrients by the summation is to better show the nutrient weights among different participants, i.e., participants with high food amounts can have high total nutrient weights. More importantly, this type of normalization actually does not affect our NR calculation, as the food distance in our NR calculation is based on the (**unweighted**) Jaccard index between foods, which means that the detailed nutrient profile was actually not included in the NR calculation.

Point 2.8: (4) In this paper, the word “personalized” is often used (e.g., L116 and other places). However, I think that this word does not seem to be rigorously defined in this study. My sense of “personalization” is that intra-individual similarity of the foods consumed over time should be larger than inter-individual

similarity of their consumed foods, but I wonder whether the authors used any measures to really quantify such tendency.

Response: We thank Reviewer #2 for this very insightful comment. Following by the reviewer's excellent suggestion, we calculated the intra-individual dissimilarity of the foods consumed over time and inter-individual dissimilarity of their consumed foods, finding that intra-individual dissimilarity of the food profiles is significantly lower than inter-individual dissimilarity of their consumed foods at both single food and food-group levels (see **Fig.R9:a-b**). In other words, food profiles are indeed highly personalized. As expected, nutrient profiles are not highly personalized (**Fig.R9:c**).

In the revised manuscript, we have added this result (see main text, page 4, lines 112-117):

“Moreover, the food profiles are highly personalized⁴⁴, i.e., the intra-individual dissimilarity of the foods consumed over time is significantly lower than the inter-individual dissimilarity of their consumed foods at both single food (**Fig.S1a**) and food-group (**Fig.S1b**) level. By contrast, the nutrient profiles, as expected, were highly conserved across different individuals and over the whole study time period for all five studies (**Fig.1:a2-e2**), and were not highly personalized (**Fig.S1c**).”

Point 2.9: (5) Overall, I believe that the authors need to provide more specific examples of their findings, to make the readers more convinced. For example, in L119-120 “The most abundant food items of individuals in most studies were from the sweetened beverages, and vegetables.” Are the vegetables really one of the most abundant food items in their diet data? (the subjects in these cohorts may mainly have practiced so-called Western diets, though?)

Response: We thank Reviewer #2 for this comment. In the previous version of our manuscript, the moisture content of each food was not excluded. Following Reviewer #2's excellent suggestion, in the revised manuscript, we have excluded the water gram in the calculating of relative abundance. We found that the most abundant food groups were grain/bread products, meat, sweets.

Point 2.10: (6) The authors performed some statistical tests to examine the significance of their results, but sometimes they took t-test, sometimes U-test, without the explanation of the choices of these methods. For example, is there any normality in their data to justify their use of t-test or z-test? If yes, why did they then use U-test (free of normality condition) sometimes?

Response: We thank Reviewer #2 for this critical comment. We apologize for the confusion. In the revised manuscript, after checking the normality of data, we have replaced U-test with t-test, finding that this doesn't affect our conclusion (see main text Fig.6 and SI Fig.S6).

Point 2.11: (7) It is still unclear to me “how” the authors controlled for possible confounding effects in their analysis, although they mentioned that they controlled for them.

Response: We thank Reviewer #2 for this comment. In this study, we used the age (months)- and calendar year-stratified Cox proportional-hazard model to compute the hazard ratios and 95% confidence intervals (CIs) of the disease according to tertiles of the NR for NHS participants from 1984 to 2014, and HPFS participants from 1986 to 2016.

In the revised manuscript (see main text, page 10, lines 344-347), we mentioned the above point. Moreover, we mentioned the following details on adjusting confounding effects in multivariable-adjusted models (see the caption of Table 1 and Table 2, main text, pages 25-26):

“Multivariable-adjusted model adjusted for age (years), ethnicity (white, African American, Asian, others), body mass index (<21.0, 21.0-22.9, 23.0-24.9, 25.0-26.9, 27.0-29.9, 30.0-32.9, 33.0-34.9, or ≥ 35.0 kg/m²), smoking status (never smoked, past smoker, currently smoke 1-14 cigarettes per day, 15-24 cigarettes per day, or ≥ 25 cigarettes per day), alcohol intake (0, 0.1-4.9, 5.0-9.9, 10.0-14.9, 15.0-29.9, and ≥ 30.0 g/d), hypertension (yes, no), hypercholesterinemia (yes, no), multivitamin use (yes, no), physical activity (quintiles), alternative healthy eating index, family history of diabetes. In NHS, postmenopausal hormone use (never, former, or current hormone use, or missing) and oral contraceptive use were additionally adjusted.”

Point 2.8: 3. In L146, I suppose that n_i and n_j in the Rao’s quadratic entropy Q shall be corrected as f_i and f_j . Also, the font sizes in Fig. 3 need to be increased for better visual inspection.

Response: We thank Reviewer #2 for those comments. We have revised them accordingly.

Finally, we thank Reviewer #2 again for reviewing our manuscript. We hope our responses have addressed her/his comments in a satisfactory manner.

Figure R1: Nutrient redundancy serves as a potential metric to predict healthy aging on NHS. **a**, Pearson correlation between the nutritional redundancy (NR) and existing healthy diet scores: Alternate Mediterranean Diet Score (AMED); Alternate Healthy Eating Index 2010 (AHEI-2010); Dash Style Diet Score (DASH); and Healthy Eating Index 2005 (HEI-2005). P values were calculated from t-test. **b**: Pearson correlation between NR and several host factors: body-mass index (BMI); education level; median income; energy intake level; Aspirin use; pack-years of smoking; physical activity. P values were calculated from two-sided t-test. **c**: Error rate of random forest classifier in the prediction of healthy aging status. **d**: AUROC of random forest classifier in prediction of healthy aging status. The participants are randomly spitted into 80% as the training set and the remaining 20% as the test set. The boxplot represents the performances of 200 independent splits. Boxes indicate the interquartile range between the first and third quartiles with the central mark inside each box indicating the median. Whiskers extend to the lowest and highest values within 1.5 times the interquartile range. All FDR-corrected P values were found using the two-sided t-test.

Figure R2: Topological features of the food-nutrient network and the human dietary pattern contributes to the nutritional redundancy. **a1-a5**, The box plots of the nutritional redundancy were calculated from the real FNN (black box), as well as the randomized FNNs (colored boxes) using four different randomization schemes: Complete randomization (Null-FNN-1); Food-degree preserving randomization (Null-FNN-2); Nutrient-degree preserving randomization (Null-FNN-3); Food- and nutrient-degree preserving randomization (Null-FNN-4). Here the degree of a nutrient is the number of foods that contain it, and the degree of a food is the number of nutrients contained in it. **b1-b5**, The box plots of NR were calculated from the real dietary records (black box), as well as the randomized dietary records (colored boxes) using three different randomization schemes: Randomized food assemblage generated by randomly choosing the same number of food items from the food pool but keeping the food profile unchanged (Null-composition-1); Randomized food abundance profiles through random permutation of non-zero abundance for each participant across different foods (Null-composition-2); Randomized food abundance profiles through random permutation of non-zero abundance for each food across different participants (Null-composition-3). Boxes indicate the interquartile range between the first and third quartiles with the central mark inside each box indicating the median. Whiskers extend to the lowest and highest values within 1.5 times the interquartile range.

Table R1. Hazard ratios (95% confidence intervals) of type 2 diabetes according to tertiles of NR in the Nurses' Health Study (NHS, 1984-2014) and Health Professionals Follow-Up Study (HPFS, 1986-2016).

NHS	T1	T2	T3	P for trend ¹
Cases/Person-year	1,436/312,411	1,245/312,509	1,133/312,388	
Age-adjusted model	1 (reference)	0.86(0.80,0.93)	0.78(0.72,0.85)	<0.001
Multivariable-adjusted model²	1 (reference)	0.93(0.86,1.01)	0.93(0.85,1.00)	0.09
HPFS				
Cases/Person-year	684/147,746	502/148,000	502/148,085	
Age-adjusted model	1 (reference)	0.73(0.65,0.82)	0.73(0.65,0.82)	<0.001
Multivariable-adjusted model²	1 (reference)	0.77(0.69,0.87)	0.82(0.73,0.93)	0.002

¹P for trend was calculated using the median value of each tertiles.

²Multivariable-adjusted model adjusted for age (years), ethnicity (white, African American, Asian, others), body mass index (<21.0, 21.0-22.9, 23.0-24.9, 25.0-26.9, 27.0-29.9, 30.0-32.9, 33.0-34.9, or ≥35.0 kg/m²), smoking status (never smoked, past smoker, currently smoke 1-14 cigarettes per day, 15-24 cigarettes per day, or ≥25 cigarettes per day), alcohol intake (0, 0.1-4.9, 5.0-9.9, 10.0-14.9, 15.0-29.9, and ≥30.0 g/d), hypertension (yes, no), hypercholesterinemia (yes, no), multivitamin use (yes, no), physical activity (quintiles), alternative healthy eating index, family history of diabetes. In NHS, postmenopausal hormone use (never, former, or current hormone use, or missing) and oral contraceptive use were additionally adjusted.

Table R2. Hazard ratios (95% confidence intervals) of cardiovascular disease according to tertiles of NR in the Nurses' Health Study (NHS, 1984-2014) and Health Professionals Follow-Up Study (HPFS, 1986-2016).

	T1	T2	T3	P for trend ¹
NHS				
Cases/Person-year	1,477/325,684	1,326/325,688	1,250/325,419	
Age-adjusted model	1 (reference)	0.92(0.85,0.99)	0.86(0.79,0.92)	<0.001
Multivariable-adjusted model²	1 (reference)	0.94(0.87,1.02)	0.90(0.83,0.97)	0.006
HPFS				
Cases/Person-year	1,358/145,486	1,285/145,662	1,290/145,726	
Age-adjusted model	1 (reference)	0.94(0.87,1.02)	0.89(0.83,0.96)	0.004
Multivariable-adjusted model²	1 (reference)	0.97(0.89,1.04)	0.92(0.85,1.00)	0.04

¹P for trend was calculated using the median value of each tertiles.

²Multivariable-adjusted model adjusted for age (years), ethnicity (white, African American, Asian, others), body mass index (<21.0, 21.0-22.9, 23.0-24.9, 25.0-26.9, 27.0-29.9, 30.0-32.9, 33.0-34.9, or ≥35.0 kg/m²), smoking status (never smoked, past smoker, currently smoke 1-14 cigarettes per day, 15-24 cigarettes per day, or ≥25 cigarettes per day), alcohol intake (0, 0.1-4.9, 5.0-9.9, 10.0-14.9, 15.0-29.9, and ≥30.0 g/d), hypertension (yes, no), hypercholesterinemia (yes, no), multivitamin use (yes, no), physical activity (quintiles), alternative healthy eating index, family history of myocardial infarction. In NHS, postmenopausal hormone use (never, former, or current hormone use, or missing) and oral contraceptive use were additionally adjusted.

Table R3. Hazard ratios (95% confidence intervals) of type 2 diabetes according to tertiles of FD in the Nurses' Health Study (NHS, 1984-2014) and Health Professionals Follow-Up Study (HPFS, 1986-2016).

NHS	T1	T2	T3	P for trend¹
Cases/Person-year	1,300/312,451	1,279/312,504	1,235/312,353	
Age-adjusted model	1 (reference)	0.98(0.91,1.06)	0.94(0.87,1.02)	0.16
Multivariable-adjusted model²	1 (reference)	1.04(0.96,1.12)	1.01(0.93,1.09)	0.85
HPFS				
Cases/Person-year	622/147,862	526/147,989	540/147,980	
Age-adjusted model	1 (reference)	0.84(0.75,0.95)	0.88(0.78,0.98)	0.01
Multivariable-adjusted model²	1 (reference)	0.91(0.81,1.02)	0.99(0.88,1.12)	0.60

¹P for trend was calculated using the median value of each tertiles.

²Multivariable-adjusted model adjusted for age (years), ethnicity (white, African American, Asian, others), body mass index (<21.0, 21.0-22.9, 23.0-24.9, 25.0-26.9, 27.0-29.9, 30.0-32.9, 33.0-34.9, or ≥35.0 kg/m²), smoking status (never smoked, past smoker, currently smoke 1-14 cigarettes per day, 15-24 cigarettes per day, or ≥25 cigarettes per day), alcohol intake (0, 0.1-4.9, 5.0-9.9, 10.0-14.9, 15.0-29.9, and ≥30.0 g/d), hypertension (yes, no), hypercholesterinemia (yes, no), multivitamin use (yes, no), physical activity (quintiles), alternative healthy eating index, family history of diabetes. In NHS, postmenopausal hormone use (never, former, or current hormone use, or missing) and oral contraceptive use were additionally adjusted.

Table R4. Hazard ratios (95% confidence intervals) of cardiovascular disease according to tertiles of FD in the Nurses' Health Study (NHS, 1984-2014) and Health Professionals Follow-Up Study (HPFS, 1986-2016).

	T1	T2	T3	P for trend ¹
NHS				
Cases/Person-year	1,438/325,560	1,339/325,704	1,274/325,527	
Age-adjusted model	1 (reference)	0.95(0.88,1.02)	0.92(0.85,0.99)	0.03
Multivariable-adjusted model²	1 (reference)	0.98(0.91,1.06)	0.97(0.90,1.05)	0.48
HPFS				
Cases/Person-year	1,374/145,508	1,265/145,700	1,294/145,667	
Age-adjusted model	1 (reference)	0.92(0.85,0.99)	0.92(0.86,1.00)	0.03
Multivariable-adjusted model²	1 (reference)	0.95(0.88,1.02)	0.97(0.89,1.05)	0.34

¹P for trend was calculated using the median value of each tertiles.

²Multivariable-adjusted model adjusted for age (years), ethnicity (white, African American, Asian, others), body mass index (<21.0, 21.0-22.9, 23.0-24.9, 25.0-26.9, 27.0-29.9, 30.0-32.9, 33.0-34.9, or ≥35.0 kg/m²), smoking status (never smoked, past smoker, currently smoke 1-14 cigarettes per day, 15-24 cigarettes per day, or ≥25 cigarettes per day), alcohol intake (0, 0.1-4.9, 5.0-9.9, 10.0-14.9, 15.0-29.9, and ≥30.0 g/d), hypertension (yes, no), hypercholesterinemia (yes, no), multivitamin use (yes, no), physical activity (quintiles), alternative healthy eating index, family history of myocardial infarction. In NHS, postmenopausal hormone use (never, former, or current hormone use, or missing) and oral contraceptive use were additionally adjusted.

Figure R3: Nutrient redundancy serves as a potential metric to predict healthy aging in HPFS. a: Error rate of random forest classifier in the prediction of healthy aging status. **b:** AUROC of random forest classifier in prediction of healthy aging status. The participants are randomly spitted into 80% as the training set and the remaining 20% as the test set. The boxplot represents the performances of 200 independent splits. Boxes indicate the interquartile range between the first and third quartiles with the central mark inside each box indicating the median. Whiskers extend to the lowest and highest values within 1.5 times the interquartile range.

Figure R4: Spearman correlation between the food diversity (FD) and nutritional redundancy (NR) (or the normalized nutritional redundancy: nNR). DAMS (dietary intake data collected using ASA24 dietary assessment tool daily over 17 consecutive days); WLVS (with four ASA24 records within one year); MLVS (with four ASA24 records within one year); NHS (with FFQ administered every four years and with total eight time points); HPFS (with FFQ administered every four years and with total seven time points).

Figure R5: Square root of median relative abundance of each food group for each tertiles group (T1-T3) based on NR of participants in NHS (a) and HPFS (b). Food group was defined in Harvard food nutrition database and food groups presented in less than 3 tertiles groups were not shown.

Figure R6: Raw food-Nutrient networks (FNN) constructed from the Harvard food composition database. We organized this matrix using the Nestedness Temperature Calculator to emphasize its nested structure⁶.

Figure R7: Food-Nutrient networks (FNN) constructed from the FNDDS database after removing those nutrients that are not specific enough to have a SIMLES or InChIKey ID, e.g., sugar, total fat, protein, total fiber, etc. We organized this matrix using the Nestedness Temperature Calculator to emphasize its highly nested structure⁶.

Figure R8: Removing fatty acids does not significantly alter NR. For participants in DMAS (a) and NHS (b), we calculated the Pearson correlation between their NR values calculated from the original food nutrient network (FNN) and the FNN with fatty acids-related nutrients (total fatty acids of saturated, total fatty acids of monounsaturated, and total fatty acids of polyunsaturated) removed.

Figure R9: Nutritional profiles are highly conserved across individuals while food profiles are highly personalized. The Bray-Curtis dissimilarity (column-1), rJSD (rooted Jensen-Shannon divergence, column-2), Yue-Clayton distance (column-3) and 1-Spearman correlation (column-4) between the food profiles of the same individuals but different time points (intra-individual) and food profiles among different individuals (inter-individual) at single food level (a1-a4) and nine major food groups level (b1-b4) and nutrient profiles (c1-c4). The Bray-Curtis dissimilarity (column-1), rJSD (column-2), Yue-Clayton distance (column-3) and 1-Spearman correlation (column-4) between the food (or nutritional) profiles of different individuals and different time points at the single food level (d1-d4) and nine major food group level (e1-e4). The Bray-Curtis dissimilarity between a pair of individuals, j and k is defined as: $BC_{jk} \equiv \frac{\sum_i |X_{ij} - X_{ik}|}{\sum_i (X_{ij} + X_{ik})}$. The rJSD dissimilarity is defined as: $D_{rJSD}(j, k) \equiv \left[\frac{D_{KL}(j, m) + D_{KL}(k, m)}{2} \right]^{1/2}$, in which $m \equiv \frac{j+m}{2}$ and $D_{KL}(j, k) \equiv \sum_{i \in S} X_{ij} \log \frac{X_{ij}}{X_{ik}}$

divergence between j and k . The Yue-Clayton dissimilarity is defined as: $YC_{jk} \equiv \frac{\sum_i X_{ij}X_{ik}}{\sum_i \sum_i (X_{ij}-X_{ik})^2 + \sum_i (X_{ij}*X_{ik})}$. In all dissimilarity definitions, X_{ij} represents the relative abundance of food/nutrient i in individual j . We only choose 100 participants in NHS and HPFS due to computational complexity.

Reference

1. Zheng, Y. et al. Associations of Weight Gain From Early to Middle Adulthood With Major Health Outcomes Later in Life. *JAMA* 318, 255 (2017).
2. Bascompte, J., Jordano, P., Melián, C. J. & Olesen, J. M. The nested assembly of plant–animal mutualistic networks. *Proceedings of the National Academy of Sciences* **100**, 9383–9387 (2003).
3. Payrató-Borràs, C., Hernández, L. & Moreno, Y. Breaking the Spell of Nestedness: The Entropic Origin of Nestedness in Mutualistic Systems. *Phys. Rev. X* **9**, 031024 (2019).
4. Tian, L. et al. Deciphering functional redundancy in the human microbiome. *Nature communications* **11**, 1–11 (2020).
5. Schakel, S. F., Buzzard, I. M. & Gebhardt, S. E. Procedures for Estimating Nutrient Values for Food Composition Databases. *Journal of Food Composition and Analysis* **10**, 102–114 (1997).
6. Atmar, W. & Patterson, B. D. The measure of order and disorder in the distribution of species in fragmented habitat. *Oecologia* **96**, 373–382 (1993).
7. Beckett, S. J. Improved community detection in weighted bipartite networks. *R. Soc. open sci.* **3**, 140536 (2016).

Reviewer comments, second round -

Reviewer #1 (Remarks to the Author):

The authors have addressed all issues I brought up in an excellent way. I have no further suggestions.

Reviewer #2 (Remarks to the Author):

The authors have extensively addressed my previous comments and I really appreciate this effort and the significant improvement in their manuscript.

Just one minor issue is about the authors' answer in Point 2.3 in their rebuttal letter: "We have tried 4 basic null models of the bipartite graph. Systematically exploring other null models of the bipartite graph (or its unipartite presentations) deserves dedicated effort, which is beyond the scope of the current study."

Regarding this issue, if it is too much to generate and test the null model for destroying nestedness while preserving the clustering, the authors may be able to at least discuss in the manuscript the current limitation of this work, which cannot entirely exclude the possibility of the presence of the other more simpler (and thus central) factor than the nestedness itself.

Relatedly, although not of the major issue in this context, the authors noted in Point 2.2 that "We found that the modularity Q of the FNN is 0.11, which is much lower than that of other bipartite graphs (e.g., plant-pollinator networks typically with $Q > 0.5$)". However, I guess that a possibly very dense and abundant total links in FNN (if yes) may result in such apparently low modularity Q , rather than this is the reflection of the truly weakly modular structure of FNN.

Response to Reviewer #1

Point 1.0: The authors have addressed all issues I brought up in an excellent way. I have no further suggestions.

We thank Reviewer #1 for reviewing our manuscript again. We are glad to know that s/he is happy with our revisions.

Response to Reviewer #2

Point 2.0: The authors have extensively addressed my previous comments and I really appreciate this effort and the significant improvement in their manuscript.

We thank Reviewer #2 for reviewing our manuscript again, and her/his very positive assessment on the revised manuscript. Next, we address each of her/his remaining comments in order.

Point 2.1: Just one minor issue is about the authors' answer in Point 2.3 in their rebuttal letter: "We have tried 4 basic null models of the bipartite graph. Systematically exploring other null models of the bipartite graph (or its unipartite presentations) deserves dedicated effort, which is beyond the scope of the current study." Regarding this issue, if it is too much to generate and test the null model for destroying nestedness while preserving the clustering, the authors may be able to at least discuss in the manuscript the current limitation of this work, which cannot entirely exclude the possibility of the presence of the other more simpler (and thus central) factor than the nestedness itself.

Response: we thank Reviewer #2 for this excellent suggestion. In the revised manuscript, we have explicitly added the following limitation of our work to the Discussion (see main text, page 13, lines 436-441):

"Forth, although we considered four null models of the bipartite food-nutrient network, systematically exploring other null models of this bipartite network (or its unipartite presentations) deserves dedicated efforts, which is beyond the scope of the current study. Hence, our analysis cannot entirely exclude the possibility of the presence of other network characteristics that are simpler than nestedness itself and can still be used to explain the NR values observed in the human diet."

Point 2.2: Relatedly, although not of the major issue in this context, the authors noted in Point 2.2 that "We found that the modularity Q of the FNN is 0.11, which is much lower than that of other bipartite graphs (e.g., plant-pollinator networks typically with $Q > 0.5$)". However, I guess that a possibly very dense and abundant total links in FNN (if yes) may result in such apparently low modularity Q , rather than this is the reflection of the truly weakly modular structure of FNN.

Response: we thank Reviewer #2 for this comment. To directly test the reviewer's hypothesis, we computed the Q values of completely randomized FNNs (with the total number of nodes and links preserved). We found that the Q values of those completely randomized FNNs are also low: $Q = 0.053 \pm 0.00056$ (50 randomizations). This supports the reviewer's hypothesis that a very dense FNN contributes to the low Q value observed in the real FNN.

Finally, we thank Reviewer #2 again for reviewing our manuscript and her/his very insightful comments that help us significantly improve the quality of our work. We hope our responses above address her/his remaining concerns in a satisfactory manner.